# Advances in Multifunctional Bioactive Coatings for Metallic Bone Implants

**DOI:** 10.3390/ma16010183

**Published:** 2022-12-25

**Authors:** Maria P. Nikolova, Margarita D. Apostolova

**Affiliations:** 1Department of Material Science and Technology, University of Ruse “A. Kanchev”, 8 Studentska Str., 7017 Ruse, Bulgaria; 2Medical and Biological Research Lab., “Roumen Tsanev” Institute of Molecular Biology, Bulgarian Academy of Sciences, 1113 Sofia, Bulgaria

**Keywords:** metallic implants, bioactive coatings, surface modifications, coating techniques, combinatorial treatments, controlled release

## Abstract

To fix the bone in orthopedics, it is almost always necessary to use implants. Metals provide the needed physical and mechanical properties for load-bearing applications. Although widely used as biomedical materials for the replacement of hard tissue, metallic implants still confront challenges, among which the foremost is their low biocompatibility. Some of them also suffer from excessive wear, low corrosion resistance, infections and shielding stress. To address these issues, various coatings have been applied to enhance their in vitro and in vivo performance. When merged with the beneficial properties of various bio-ceramic or polymer coatings remarkable bioactive, osteogenic, antibacterial, or biodegradable composite implants can be created. In this review, bioactive and high-performance coatings for metallic bone implants are systematically reviewed and their biocompatibility is discussed. Updates in coating materials and formulations for metallic implants, as well as their production routes, have been provided. The ways of improving the bioactive coating performance by incorporating bioactive moieties such as growth factors, osteogenic factors, immunomodulatory factors, antibiotics, or other drugs that are locally released in a controlled manner have also been addressed.

## 1. Introduction

With the global aging and obesity of the population, the necessity of applying orthopedic implants has increased rapidly. For improving the quality of a patient’s life, durability of a minimum of 15–20 years for older persons and more than 20 years for younger people is expected for a bioimplant [1]. Orthopedic implants are synthetic tools designed to provide biological support to damaged tissues or organs in the living organism and restore physiological functions. Metallic materials were first used as first-generation bone implants due to their physical and mechanical properties and inert nature. Still, among all manufactured implants, about 70 to 80% are produced from bio-metallics. However, the failure of metallic implants mostly originates at the tissue–implant interface due to poor bonding leading to the formation of a non-adherent layer of movement at this interface [2]. A study of almost 338,000 procedures for knee revisions demonstrated that aseptic loosening accounted for 20.4%, infectious factors (septic loosening) for approximately 20.3% of implant failure [3]. The aseptic loosening of the implants is a slow process that develops over the years. It can be related to the release of allergens by metals, mechanical incompatibility and occurrence of stress shielding effect on bone, poor wear and corrosion resistance leading to the formation of wear debris and release of toxic corrosive ions which can trigger inflammatory reactions [4]. Early and late (more than 24 months) implant infections are caused by different pathogens (such as *Staphylococcus aureus*) that rapidly proliferate and secrete extracellular polymers to form biofilm. In the presence of tiny gaps within the bone and prosthesis interface, particles and/or pathogens can accumulate and hinder the direct contact between the graft and bone [5]. All these problems are related mostly to the surface of the metallic implant. To minimize revision surgery, numerous approaches including oral intake of antibiotics, selection of implant material, engineering of modified implants, etc., have been developed. Simultaneously, by modifying the chemistry, morphology and structure of the metal implant surface, different properties than in bulk can be obtained. Therefore, a way of preventing implant failure and loss is applying coatings that improve the mechanical properties and corrosion resistance of materials and promote osseointegration by stimulating osteoblast recruitment and differentiation to achieve bone formation, as well as removing or killing pathogens, thus preventing biofilm formation. Coatings based on stimuli-triggered therapeutic strategies with improved drag bioavailability and fast on-site bone healing effects have been also proposed [6]. In that way, the performance and service life of dental and orthopedic implants can be substantially improved. Focusing on enhancing various shortcomings of metal implant surfaces, biocompatible coatings can offer a unique combination of properties that best functionalize the implants. For that reason, this review outlines the main materials used for metal implant manufacturing and some general coating techniques and focuses on recent trends in the design and performance of biomedical coatings for metallic implants used for orthopedic and dental applications. Some techniques for the reduction of infections and advanced combinatorial treatment are also described together with the associated drawbacks of the coated systems.

## 2. Bone Regeneration

The main objective for applying a biocompatible coating is to enhance the osteoconductive, osteo-inductive and osteogenic performance of the metal implant surface. Osteo-conductivity is the ability of the coating to act as a scaffold for extracellular bone matrix formation where osteoblasts can adhere and proliferate, while osteo-inductivity refers to the ability of the surface to stimulate the differentiation of precursor (stem) cells into osteoblasts. Osteogenic coatings alloy osteoblasts to produce calcium nodules for calcification of the collagen matrix of the newly formed bone structure [7].

### 2.1. Molecular Mechanisms

Mammalian bone can be found in two forms: woven and lamellar. While woven bone is characterized by a random distribution of collagen fibrils and mineral crystals due to its fast deposition, occurring during the development and repair of fractures, lamellar bone is more precisely arranged, being formed much more slowly [8]. Consequently, woven bone is mechanically much less stable than lamellar bone, thus being replaced by lamellar bone during remodeling.

The principal role of bone is skeletal support, providing body mobility and the protection of organs. For example, the jaw bones—maxilla and mandible—hold the teeth in place and transmit chewing forces from the muscles of mastication to the teeth. Bone architecture is highly adapted to the structural needs of high strength at minimum weight. The way to accomplish this is to form dense, highly mineralized cortical bone and porous trabecular bone, also called cancellous bone. Bone is also responsible for blood production, mainly in the bone marrow and bone acts as a storage medium for minerals such as calcium and phosphates, as well as proteins such as growth factors, which are deposited in the matrix and later released to responding cells [3]. The trabecular bone is the site of hematopoiesis, involved in fat and iron storage. Ca^2+^ homeostasis also occurs in trabecular bone and hematopoietic activity is limited to the bone marrow [9].

The bone is composed of a hierarchically organized composite material of fibrous type 1 collagen, non-collagenous proteins forming the osteoid and intercalated hydroxyapatite crystals. Although bone seems relatively static, it is a living and dynamic tissue consisting of three main cell types, osteoblasts, osteoclasts and osteocytes, as well as bone lining cells, which cover the bone surface and are marginally involved in bone remodeling [10]. Usually, osteoclasts and osteoblasts travel the bone surface, whereas osteocytes are embedded in the calcified bone matrix throughout the entire bone mass. Through canals (canaliculi), osteocytes remain connected to neighboring cells, thus forming a 3D cellular network in the bone structure.

Osteoblasts originate from pluripotent mesenchymal stem cells. Apart from differentiating into osteoblasts, these cells can differentiate into chondroblasts (forming cartilage), fibroblasts, adipocytes, myoblasts and other cell types [11]. The morphology of osteoblasts is dependent on the stage of functional differentiation. They attach to the extracellular matrix and other cells via transmembrane proteins, such as integrins, connexins and cadherins, which enable them to react to metabolic and mechanical stimuli [12,13]. The main task of mature osteoblasts is to synthesize proteins, such as collagen I and other glycoproteins such as osteopontin, osteocalcin and growth factors.

Mineralization of new hydroxyapatite is initiated and regulated through the formation of matrix vesicles (MV). By supplying sufficient calcium and phosphate ions, osteoblasts release MV from the cell surface into the extracellular matrix. Further mineral accumulation in the MV is achieved through ion channels and transporters leading to initial nucleation and calcification in the MV. Finally, needle-like hydroxyapatite crystals are released from the MV into the extravesicular fluid, these crystals subsequently being incorporated into the collagen fibril network [14,15]. Some osteoblasts become incorporated into the matrix and turn into immature osteocytes embedded in the fully mineralized and matured matrix [16,17].

Today, it is known that osteocytes are essential in both mechano-sensation and mechano-transduction [18,19]. Over 80% of all cells in the bone are osteocytes, which originate from highly specialized and fully differentiated osteoblasts, as mentioned above. Any microcracks in the bone can induce osteocyte apoptosis, which activates remodeling [20]. Bone-resorbing osteoclasts and bone-forming osteoblasts at the cellular level execute bone remodeling. Osteoclasts resorb existing bone and leave behind a resorption pit. They are followed by osteoblasts building new bone through collagen deposition and mineralization of calcium phosphate. An active osteoclast can resorb around 200,000 µm^3^ per day and this compares to 15–20 days of bone formation by seven to ten generations of osteoblasts [21]. In the bone-remodeling cycle, local communication and molecular interactions between osteocytes, osteoclasts and osteoblasts play a crucial role, together with the signaling system, involving RANKL (receptor activator of nuclear factor κB ligand), PTH (parathyroid hormone) and transforming growth factors (TGFs) [22].

### 2.2. Cells and Their Microenvironment

Every cell is embedded into a microenvironment—the extracellular matrix (ECM). This matrix consists of several proteins, such as fibronectin (Fn), collagen, elastin and laminin. The other main components are glycosaminoglycans (GAGs) which normally covalently link to proteins. These macromolecules are secreted locally and assembled into an organized meshwork by the cells. The cell’s microenvironment can be divided into geometrical, mechanical and chemical components. The geometry is determined by the cell shape, the mechanics by the rigidity of the microenvironment and the chemistry by the extracellular molecules with which the cell interacts. These properties of the local cell microenvironment regulate cell behavior in concert with autocrine and paracrine soluble or matrix-bound signaling molecules [23,24].

Typical amino acid sequences of the matrix proteins mediate cell adhesion. In general, when biomaterials are functionalized with peptides of this type, the peptides are adsorbed to the surface or are covalently linked to them with the use of different chemical reactions and surface modifications. Typical sequences involved in cell adhesion are Arginine-Glycine-Aspartic acid (RGD), Pro-His-Ser-Arg-Asn (PHSRN), Lys-Arg-Ser-Arg (KRSR), Asp-Gly-Glu-Ala (DGEA), etc. [25]. The well-known RGD sequence is a domain that occurs in many matrix proteins [26]. It is recognized by a range of cell types by various types of integrin receptors [27]. The PHSRN sequence has a synergetic effect with RGD, crucial for optimal integrin binding [28]. KRSR—a heparin-binding domain of fibronectin and collagen—plays an essential role in the selective adhesion of osteoblasts. It also inhibits the adhesion of fibroblasts, endothelial and epithelial cells [29,30]. It is recognized by the non-integrin receptor-cell-membrane heparin sulfate proteoglycan. DGEA is a typical sequence for type I collagen recognized by α_2_β_1_ integrins [31]. Similarly, as was done with the bone morphogenic protein-2 collagen binding domain (BMP2-CBD), these bioactive sequences can be designed and produced with additional domains to direct their attachment to various biomaterials, such as collagen or hydroxyapatite.

Angiogenesis is another important factor needed for an appropriate bone healing procedure. New blood vessel formation is essential for supplying the cells, nutrients and oxygen and for removing waste products. Angiogenesis is a complicated process involving various growth factors, ECM molecules and cell types. Among others, osteopontin is a phosphoprotein that participates in bone metabolism, mediates inflammatory responses and plays a crucial role in angiogenesis [32]. Recently, an osteopontin-derived peptide Ser-Val-Val-Tyr-Gly-Leu-Arg (SVVYGLR) has been identified as having pro-angiogenic properties, supporting the adhesion and migration of endothelial cells [33]. Moreover, it enables bone marrow stromal cell differentiation into endothelial cells [34].

Much effort has been invested in engineering implant surfaces that mimic and contain native ECM proteins. However, an adsorbed mixture of proteins with random unfolding, orientation and conformational states presents a divergence from natural, intentionally arranged protein layers. Immobilizing entire native proteins to surfaces can provide many functions because of the various domains within the molecule-engineered cellular environments [35]. An alternative approach is using peptides instead of complete proteins. Peptides based on the primary structure of the receptor-binding domain of an entire protein, such as RGD from fibronectin, aim to target specific cellular adhesion and interactions [35].

Normal tissue cells are dependent on anchoring to a substrate for survival. However, tissue can have a wide range of stiffness, with the brain in the soft range (about 1 kPa), bone in the hard range (about 100 kPa) and many others in between. Therefore, a cell experiences different tissue stiffness in vivo. Physical properties of tissue are also altered at a wound healing site and can change during disease progression [36,37,38]. The initial cellular response to mechanical signals occurs in seconds to minutes (Figure 1). A cell in culture for several days can be expected to experience countless stimulus–response cycles. The mechanism behind mechano-sensing is complex. Focal adhesions, focal complexes, integrins and cytoskeleton have been identified to play a key role in the mechano-sensing cells’ ability [39,40]. All these processes are essential for implant integration.

The clinical success of implant treatment is based on the concept of osseointegration, first described by Brånemark [41], today defined as “the formation of a direct interface between an implant and bone, without intervening soft tissue” [42]. Osseointegration is the antithesis of the fibrous encapsulation of a foreign-body response, where an implant or any other foreign material inserted in the body is encapsulated in fibrous tissue. Describing osseointegration is not possible without highlighting the eminent role of the implant materials and their surface [43]. In addition to obvious factors, such as cleanliness and sterility of the implant surfaces, which are indispensable, surface roughness and surface energy influence osseointegration. It has been shown in vitro, in vivo and clinically that rough implant surfaces show better osseointegration properties than smooth surfaces. There are other clinical fields in which the concept of osseointegration is applied, such as orthopedics, where it is named osteointegration. These areas will not be discussed further here but are outlined elsewhere [44,45]. The role of implant materials, preparations and surface modifications are described in the following text.

## 3. Requirements and Generations of Materials Used for Bone Implants

The requirements for implant materials can be both general and specific (Figure 2). The general requirements for biomaterials are related to (a) their clinical performance—to use a metal or alloy as a biomaterial, it should be biocompatible. This means that it must not be rejected and should avoid harmful effects on the host while working as a medical therapy for a certain response in the living organism [46], and (b) manufacturing criteria, including easy fabrication with complex geometry and affordable price.

Simultaneously, the biomaterial should comply with important specific requirements that determine its performance in a particular application. As seen in Figure 2, the most important implant-specific properties can be divided into (a) bulk, (b) chemical and (c) textural properties [47]. The bulk properties can be subdivided into mechanical properties and degradability. The mechanical properties of a biomaterial define its response under the influence of various forces. These properties are critical to consider, since during healing the implant bears much ununiform stress and load [48]. To avoid stress at localized points (stress shielding effect), the modulus of elasticity of both bone and implant should not show significant differences. When the biomaterial is used as a scaffold, its mechanical properties must be boosted in line with the degradability. The degradation rate that is dependent on different chemical and physical processes such as dissolution, phase transformations, etc., should correspond to the increase in new tissue formation. In bone implants, the degradation rate should be lower than 0.5 mm per year [49].

The chemical properties of metal biomaterials are linked to their chemical activity and corrosion resistance. Various factors such as a change in oxygen level or pH value in the bio environment or diffusion of ions may contribute to corrosion. The corrosion behavior of the material is dependent on its relative crystallinity, crystalline size and textural properties. The latter ensures the interaction with the bio-environment, tissue and cells and aids the healing processes. There is much evidence that the surface characteristics of the implants affect cell attachment, proliferation, migration and differentiation [50]. Some authors [50] divided the textural factors affecting the biocompatibility into two groups: (a) surface chemical properties such as hydrophilicity, free energy, polarity, crystallinity, electrostatic interactions and mobility of the surface functional groups; (b) surface physical properties such as surface topography, roughness, density, thickness and adhesion of the layer. Theoretically, the interaction of the cell with the outermost surface layer equals about 0.1-nm thickness [51]. The micro-surface roughness is important in providing contact guidance for filopodia attachment, while nanoscale roughness influences protein absorption that aids in cell attachment to protein through the integrin [52]. The wettability influences the conformation of the absorbed proteins on the surface [50] and, together with a higher percentage of crystallinity (over 70% [53] and optimal porosity level of about 30% for titanium scaffolds [54], the biocompatibility properties of the material are enhanced. To stimulate bone ingrowth, the minimal pore size should range between 100 and 150 μm [55,56].

Still, it is hard for a single metallic material to fulfill all general and specific requirements for biomaterials for implant application. Although metals and their alloys (first generation of biomaterials) meet many of the desired properties for biomaterials, their interfacial bonding with the surrounding tissue varies from poor to virtually absent [57]. Given the improved formation of adherent layer and reduced movement at the bone-implant interface, by surface modification of this first generation of biomaterials their specific properties have been upgraded. In contrast to the first generation of bioinert materials that form fibrous tissue capsules, the second generation of biomaterials focuses on the development of bioactive (generating hydroxyapatite layer on their surface) or biodegradable (bio-absorbable in a progressive manner) materials that promote specific cellular responses [58]. The third generation of biomaterials is designed to be temporary 3D scaffolds that stimulate tissue regeneration at a molecular level, angiogenesis and nutrient supply [59]. The fourth generation of biomaterials is based on personalized interaction with tissue and cellular processes depending on four requirements: receptivity, activity, autonomy and inertia [60].

## 4. Metals and Alloys for Bone Replacement

Metals and alloys are generally used for implants where high load-bearing capacity and strength are required. Compared to other biomaterials used for hard tissue replacement such as ceramics and polymers, metals display cost effectiveness, durability, mechanical strength and appropriate corrosion resistance. However, most of the alloys release metal ions in the surrounding tissue that enters the blood circulation and, in higher concentrations, the ions accumulate in the liver and spleen which can lead to cytotoxicity and organ failure upon prolonged exposure. They also suffer from impaired tissue growth because of inadequate attachment of the implant, higher risk of infections and slower healing time, compared to ceramic biomaterials [57]. Currently used metals and alloys in orthopedic metal-based implants consist of titanium-based alloys, cobalt-based alloys, magnesium-based alloys and stainless steels.

### 4.1. Stainless Steel

Stainless steel (SS) containing about 18% Cr and 8% Ni such as 316L and 304 is the most commonly used alloy for removable orthopedic devices such as screws, pins, bone plates, medullary nails, etc., because of its low price [61]. The presence of Cr allows the formation of a strong adherent Cr_2_O_3_ layer that favors corrosion resistance. Despite easy fabrication and low-cost, stainless-steel implants suffer from the stress shielding effect (because of their high elastic modulus of around 200 GPa), the occurrence of allergic reactions due to ion release [62], and are susceptible to pitting corrosion attacks. To reduce these adverse effects, small quantities of Mo (2–4 wt%), to improve the corrosion resistance and strengthen the alloy, have been added to the steel. Additionally, Ni-free SS with high corrosion resistance and biocompatibility has been developed [63,64]. However, compared with Ti-based alloy, the density (7.9 g/cm^3^) is higher while the osteointegration, biocompatibility and corrosion resistance of SS implants are lower which causes fewer implant success rates [64].

### 4.2. Co-Cr Alloys

Compared with stainless steel, Co-based alloys are characterized by superior strength. Despite their difficult fabrication, Co-based alloys are more resilient to corrosion and wear and display better biocompatibility [65]. Alloyed with Cr and Mo, some Co-based alloys like Co-Cr-Mo and Co-Ni-Cr-Mo are specially used for implants in the hip, knee and shoulder prosthesis [66]. Compared with other metallic implants, Co-based alloys display a better elastic modulus, stiffness and high density (8.3 g/cm^3^) [67]. However, the elastic modulus and ultimate tensile strength of Co-based alloys (200–250 GPa and 400–1000 GPa, respectively) are about 10 times higher than those of human bones which can result in a stress shielding effect. Co-based alloys are also not ideal for joint and bearing surfaces because of their sub-par frictional properties [68]. Additionally, their alloying elements such as Ni, Cr and Mo were proven to be toxic when released from the implant surface in the body fluid during corrosion processes. Excessive accumulation of these elements in organs such as kidneys, liver, lungs and blood cells can trigger their damage [65].

### 4.3. Ti and Ti Alloys

Ti and its alloys have an advantage over steel and Co-based alloys because of having low density (4.5 g/cm^3^), low weight and biocompatibility. Pure Ti has the advantage of being very highly corrosion resistant because of the formation of TiO_2_. Titanium alloys show better mechanical properties in comparison to pure Ti. About 50% of the total usage of titanium alloys falls on Ti6Al4V alloy because of its excellent formability, structural stability, weight-to-strength ratio and corrosion resistance [69]. However, together with Ti6Al7Nb, Ti6Al4V alloy suffers from high friction and low wear resistance [70]. Though proven to be highly biocompatible and corrosion resistant, during long-term implantation, α+β structured Ti6Al4V and Ti6Al7Nb alloys also suffer from the release of toxic alloying elements, risk of stress shielding and high cost [71]. The presence of Al and V in the Ti alloys results in a release of toxic ions in the body milieu under the physiological environment which triggers adverse health problems [72]. Hence, scientists focus on β structured alloys containing Nb, Zr and Ta that substitute Al and V in the composition. Such alloys are characterized by good mechanical properties that can be tuned by heat treatment, better formability, lower elastic modulus (down to around 50 GPa) and higher wear and corrosion resistance at the expense of a higher price [73].

### 4.4. Mg Alloys

In contrast to the above-mentioned alloys, Mg-based alloys are used for the production of biodegradable orthopedic implants that eliminate the need to be retrieved via a second surgery. The revision surgery is discomforting and expensive for the patients and can lead to possible complications and infections [74]. Mg and its alloys are among the lightest structural materials, with densities very similar to that of human cortical bone (1.75 g/cm^3^) [75]. Additionally, Mg is an essential element in the construction of soft tissue and bone. Mg and its alloys such as Mg-Ca-Zn, Mg-Zn-Ca-Mn with enhanced mechanical and corrosion properties have been applied for making bone screws and pins, wires, sutures, surgical clips, etc. [76]. Except for low density and lightweight, Mg-based alloys have excellent mechanical properties, elastic modulus close to that of bone and good biocompatibility [77]. A challenge in using Mg and its alloys is their high degradation rate in vivo via electrochemical mechanisms of dissolution. Besides, a decrease in mechanical properties due to corrosion, alkalization and hydrogen generation in the surrounding tissue is observed which can inhibit implant tissue interaction and promote tissue necrosis [78]. To overcome this shortcoming, scientists develop coatings or surface treatment procedures to postpone the start of degradation [79].

A comparison of some mechanical properties of the main metallic materials used for orthopedic implants is presented in Figure 3.

Although metallic implants are classified as the first generation of biomaterials, they evolve and are currently in large-scale use. Because of the wide variety of surface treatments modifying their roughness, wettability, topography, etc., the quality of bone-to-implant anchorage can be substantially improved.

## 5. Coating Methods

For the successful modification and functionalization of metallic biomaterials, it is necessary to develop techniques for changing their surface composition, morphology and structure without compromising their mechanical properties. Various coating materials such as metals, alloys, ceramics, glass, polymers, organic/inorganic hybrids, biomolecules and composites have been used to achieve bioactivity. The deposition process has to offer a high degree of flexibility and sufficient adhesion to the substrate, since it has to meet various demands for specific biomedical applications. To produce the coatings, high energy is applied to the system to achieve deposition through melting or vaporization and subsequent solidification or accelerated flow of particles toward the metallic substrate [80]. The resultant structure of the coatings can be divided into amorphous and polycrystalline, depending on the preparation conditions and the deposited material. Since the structure of the bone is microcrystalline while the amorphous phases are more prone to dissolution, the stable crystalline phases are preferred for biomedical applications.

Presently, coatings can be fabricated in various ways. The techniques can be divided into chemical, thermochemical, electrochemical, physical, electrophysical and 3D printing. The incorporation of ingredients by micro- and nanoencapsulation techniques are also promising options for controlled drug delivery.

### 5.1. Chemical Methods

The sol-gel method is usually used for the synthesis of oxide materials. It includes the preparation of precursor solution, deposition of “sol” (formation of a colloid suspension) onto the substrate and heat treatment for densification or “gel” (viscous or solid material) formation. The precursor materials can be both inorganic salts in an aqueous solution or metal alkoxides in organic solvents. The deposition of the sol is carried out via electrodeposition, spin- or dip-coating techniques (Figure 4). The technique is frequently used because of its simplicity, cost-effectiveness and ease of incorporation of active ingredients (such as growth factors recombinant human bone morphogenetic protein-2 (rhBMP2) [81] and ability to adjust coatings with different thicknesses to substrates with various shapes [82]. The abundance of chemical and physical parameters that influence the structural properties of the coatings renders the possibility of effective control over their structure and homogeneity. The obtained coatings can be porous which increases the surface area in contact with tissue and fluids, but the adhesion ability of the coatings to the metal surfaces is low due to thermal mismatch. To enhance adhesion as well as density, sol-gel coatings can be sintered at high temperatures (more than 1000 °C) or an intermediate layer can be formed [83,84]. Because of the long processing time, the need for post-sintering and the cracking of the final coating due to shrinkage, the application of the sol-gel route is limited in industry.

Sol-gel techniques are also used for the production of hydrogels and aerogels with a large specific area and high porosity at moderate temperatures. Hydrogels are formed from hydrophilic polymers such as polyvinyl alcohol (PVA), polyacrylic acid (PAA), polyglycols, hyaluronic acid, collagen, gelatin and dextran, with elastic and swelling properties, while aerogels are obtained when wet gels (usually silica based) are super-critically dried and the liquid phase is replaced by a gaseous phase. Thus, aerogels with high porosity, low density and thermal conductivity are formed [85]. Polysaccharides such as methoxyl pectin and xanthan incorporating indomethacin, are also used to prepare aerogel coatings on SS substrate after supercritical CO_2_ drying [86]. However, hydrogels have low adhesion to a metal surface which imposes physical or chemical treatment that bonds or crosslinks the hydrogel to the surface by chemical or physical bonds [87].

The electroless deposition method, also known as plating, chemical bath deposition, solution growth technique, or controlled precipitation, is mostly used to prepare metal oxide films at lower temperatures. During the process, metal ions are complexed by ligands by controlling the pH and then the substrate is immersed under a temperature range of 60 to 100 °C to form, usually, metal hydroxide films. After that, the hydroxide film is transferred to the oxide by annealing. This technique is simple, unexpensive and suitable for complex geometries. The process is used for the deposition of the biodegradable and biocompatible iron coating on pure Mg to increase its corrosion resistance [88], as well as to produce antibacterial ZnO and hybrid ZnO/Ag on biodegradable Mg-Ca alloy [89]. Although reliable, electroless deposition is a lengthy process that requires the chemical bath to be replenished often, while the coatings may show imperfections such as microcavities with columnar structure and pitting marks [90].

Employing heterogeneous nucleation and crystal growth of coating, biomimetic deposition of films with bonelike properties can be obtained [91]. The coatings are prepared, initially pretreated either with acidic or alkaline solution metallic substrates by immersion in simulated body fluid (SBF) at body temperature (37 °C) and physiological pH (7.4). After several weeks, CaP-based coatings with different Ca/P ration and crystallinity on the substrates are deposited [92]. This method allows additional bioactive components such as osteogenic agents to be co-precipitated and incorporated into the structure of the coating [93]. To enhance the mechanical properties, carbon nanotubes can be incorporated into the coating structure by adding them to the SBF solution [94]. However, this method requires a long processing time and maintenance of ion concertation, while the coatings suffer from poor bonding strength.

The wet-chemical methods also include common techniques for covalent bonding of target molecules onto metal (titanium) surfaces using phosphonic acids [95] or catechol [96] to create organofunctional anchors to graft bioactive polymers with different architectures. This kind of bonding counts on the covalent connection of end-functionalized polymers that polymerize in situ, initiated from the surface. Thus, demanding diffusion of monomers towards the propagating radicals is required. For that reason, Chouirfa and co-authors developed a technique based on catechol surface modification designed to enable bioactive ionic polymers bearing thiol end groups to be attached using a thiolene click reaction [96].

To produce coatings with high purity and crystallinity at low setting-up costs, spray pyrolysis techniques can be used. They employ processes held at atmospheric pressure in the air at ambient conditions. The setup consists of generators (such as ultrasonic, electrospray, centrifugal, pressure and pneumatic atomization) that produce mist from a spraying solution containing the precursor material [97]. The mist is transported to the surface of the substrate which is heated to achieve pyrolysis of the solution, thus forming a solid coating on the top. The synthesis of nano-coatings can be flame-, plasma-, microwave-, or laser-assisted [98]. However, post-deposition annealing may be necessary to obtain the required coating characteristics [99]. Depending on the solvent and deposition parameters, solid or molten metal moieties may vaporize upon approaching the substrate and then the coating may grow via a true chemical vapor deposition (CVD) reaction. Recently, the method has been used for the production of Zirconia-incorporated bioactive glass films on pure Ti substrates [100], thin oxide films [101] and ultra-porous HAp network on Ti alloy discs [102] by using flame spray pyrolysis. It is a one-range synthesis process with a high scalability and production rate that is capable of producing coatings of a wide range of material compositions on nonplanar implants.

### 5.2. Thermochemical Methods

Thermal spraying is a process where molten or semi-molten micrometer-sized particles are sprayed onto a substrate at a high speed. Depending on the heating source, the spraying process can be divided into electric arc, flame, plasma, kinetic spraying, etc. while the process can be held both at atmospheric pressure or in a vacuum. The principal of coating deposition by using a plasma spraying technique is illustrated in Figure 5. Thermal spraying is a favorable process to deposit bio-ceramics, bio-glasses [103], polymer [104] or composite coatings such as hydroxyapatite (HAp)/Ti [105] on metal implant materials. Due to the high impact velocity, the coating material particles are flattened in the form of lamellae or splats. Although the process is cost-effective with a high deposition rate and the coatings’ surfaces are micro rough, the obtained films are usually porous, inhomogeneous, amorphous with low adhesion strength, and prone to coating spallation due to compressive residual stresses [80]. Though the increased surface roughness is considered as an enhancing factor in the implant–bone interaction, the roughness should be in a certain range. The process is less suitable for temperature-sensitive materials, biological molecules and drug release applications due to the high process temperatures. Additionally, mechanically bonded coatings with different bond strengths are poorly accepted for biomedical applications [93]. To minimize the above-mentioned disadvantages of the high-temperature thermal processes, cold spraying, where the particles are heated below the melting temperature and accelerated by carrier gas toward the substrate, has also been conducted for biomedical applications [106]. The heated particles participate in the successful deposition of coatings such as HAp/graphene [107] or Ta [108] on metal substrates while retaining their crystal structure even after deposition.

Another high-temperature technique is hot isostatic pressing in which high isostatic pressure at high temperature is used to obtain dense ceramic coatings. The application of uniform pressure all over the system allows for removing the shape limitations of the substrate. For instance, this method is used for the deposition of HAp coating at a relatively low temperature (135 °C) and density close to that of bone [109].

The hydrothermal crystallization method is a single- or multistep process for coating deposition from solution onto a metal substrate at different temperatures (up to about 200 °C), various pH and high autogenous pressure in a hydrothermal reactor. During the single-step process, due to heating, the pressure increases in the reactor and coatings deposit on the sample surface in a supercritical environment. The multistep process is applied to already existing coatings to form a targeted phase. A single-step deposition method of the CaP layer is used for improving the corrosion properties of Mg-based alloys [110], while a multistep process was applied for the deposition of HAp on either Mg alloy or Ti6Al4V [111]. Due to its simplicity and cost-effectiveness, the hydrothermal method is applied in mass production [112].

Chemical vapor deposition (CVD) methods form films through a chemical reaction between the incoming species on the metallic surface. The chemical reaction that occurs during the CVD processes depends on the volatile precursor transported via the vapor phase and the by-products that together form a non-volatile solid through a chemical reaction. Liquid or gas source materials are used during the various variations of the CVD methods such as plasma-enhanced CVD (PECVD), metal-organo-chemical vapor deposition (MOCVD), low-pressure CVD (LPCVD), pulsed CVD, spray, chemical bath, etc. [113]. The energy used to drive the chemical reaction can be laser, photon, or temperature. Differently structured films (amorphous, polycrystalline, epitaxial and uniaxial) with a high degree of purity can be deposited. The main disadvantage is related to the limited components that are volatile at nearly ambient temperatures, the generation of toxic by-products, or the use of toxic precursors [114].

Pulsed CVD, also known as atomic layer deposition (ALD), is based on self-limiting reactions between two gaseous precursors and allows for the deposition of thin oxide films layer-by-layer at relatively low temperatures and moderate pressure. The ALD method was used for the deposition of amorphous and homogenous TiO_2_ layers [115] or to enhance the functionality of nanotubular Ti surfaces [116]. The thermodynamic complex processes that occur during CVD depend on temperature, pressure, flow rate concentration of chemical species and reactor geometry [117]. The process is resourceful for producing simple and complex compounds in the form of coatings with uniform thickness and controlled properties, but requires expensive equipment.

### 5.3. Electrochemical Methods

Electrochemical methods produce good quality films at a low price without the need for expensive equipment, while the waste is limited to the solution. They depend on the solution composition, pH value, viscosity, etc.

By electrophoretic deposition (EPD), an electric field is applied between two electrodes and various charged particles, such as fine powders, colloids, or macromolecules that are dispersed or suspended in the liquid, move toward the oppositely charged electrode thus leading to accumulation (by coagulation and precipitation) on the deposition electrode (substrate) (Figure 6). This technique can be employed in the production of homogenous bio-ceramic HAp films [118] on metallic substrates as well as polyacrylic acid films containing nanotubes and oxide particles [119]. The process requires a short process time and gives the ability to produce interconnected porous coatings with controlled thicknesses up to 2 mm [120]. The EPD technique also allows the incorporation of bioactive ingredients such as antibiotics [121] and zoledronate (medication used to treat several bone diseases) [122]. The main issues are related to the coating adhesion, occurrence of cracks and difficult scalability [123].

During electrochemical deposition, the substrate is placed as a cathode at either constant or pulse current/voltage. The latter gives more possibilities of controlling the properties of the coating by the process parameters. Good shape conformity, uniform coating thickness, short processing time and low processing temperature (usually room temperature) are the main advantages of electroplating techniques. The method allows the synthesis of metallic [124], ceramic [125], polymer [126] and composite [127] coatings. The major disadvantages include inadequate bonding to the surface and the development of stresses in the coatings [128]. The critical factor for adhesion is surface roughness. For ceramics that cannot dissociate from ions, their deposition on the metal base is difficult and the obtained surfaces are often non-homogeneous and non-stoichiometric [129].

Besides the implementation of coatings, another electrochemical strategy is changing the topography by forming a passive layer on the surface when exposed to oxygen. One of the most effective techniques is anodization. Thus, self-assembled titania nanotubes (TNTs) can be fabricated on the surface of Ti and its alloys. By varying the applied potential, time, electrolyte temperature, pH, concentration, etc., the characteristics (diameter size, branched tube, double-walled, double-layered, etc.) of TNTs can change [130] which are known to affect osteoblast cell adhesion and proliferation [131]. The TNTs can be tailored with drug-loaded properties because of their open-ended tubular shape. The loading procedure can be achieved by different conjugation strategies (for growth factors such as bone morphogenic protein-2 (BMP-2) [132] or using vacuum-freezing technologies for drugs such as antibiotics [133]. Despite these advantages, the mechanical stability of TNTs under load-bearing conditions as well as their cytotoxicity caused by the released ions or debris is still under consideration.

Among the electrodeposition methods, plasma electrochemical oxidation (PEO), also called micro-arc oxidation (MAO), is a technique developed on anodic oxidation that brings out a rough surface rich in biological elements due to high-voltage oxidation in electrolytes containing different compounds. The formation of MAO coating occurs both due to the oxidation of the substrate and the incorporation of compounds or particles from the electrolyte (Figure 7). It is a hybrid of conventional electrolysis and an atmospheric plasma process to produce well-adherent coatings on different valve metals. The advantage of MAO is that it can produce highly microporous oxide layers on implant surfaces, which is important for anchoring the bone [134]. The electrolyte may be silicate, phosphate, aluminate, sodium tetraborate, or phytic acid to produce various coatings. For example, using calcium acetate and sodium phosphate crystalline TiO_2_/Hap, coatings on Ti substrates have been prepared [135], though HAp produced by MAO is usually of low crystallinity. The HAp crystallinity and surface roughness can be enhanced by increasing the voltage and duration of the MOA process, which subsequently triggers the appearance of cracks and failures within the coating. An alternative method to increase crystallinity is to apply subsequent hydrothermal treatment [136], ultrasonic assistance [137] or immersion in SBF [134]. The MAO process is stable and reliable, with good reputability and environmental friendliness [138]. Besides, owing to its ability to form highly adherent microporous coatings, its simplicity and its cost-effectiveness, MAO is becoming an indispensable technique for the generation of functional nanostructured coatings. However, the pores can serve as channels for corrosive electrolyte ingression. Besides, in the most common Ti alloy (Ti6Al4V), a large amount of mixed oxide aluminum titanate is produced [139].

### 5.4. Physical and Electrophysical Methods

The deposition process during the PVD processes takes place in a vacuum chamber. The deposition of coatings by gas-phase processes such as evaporation and sputtering produces a thin film with good purity and structural properties. The transport of free species emitted by the target and accelerated towards the substrate allow controlling of their electric charge and kinetic energy. PVD techniques are environmentally friendly, the coatings have superior mechanical biocompatible properties and they can be applied to temperature-sensitive substrates [140]. However, these methods suffer from some drawbacks such as high processing and equipment costs, the need for a large number of material targets, gaseous waste treatment, strict instrumentation requirements and the inability of drug release applications.

Vacuum thermal evaporation is a simple PVD technique that is usually used for obtaining amorphous thin films. The material is resistively heated below 1500 °C in a vacuum condition until vapor pressure is produced. During the process of evaporation of the covering material, a potential difference to the substrate is applied at a medium or high-vacuum level. Commonly evaporated films of Ti, TiO_2_ and calcium phosphate are used to tailor the chemical and topographical properties of biomedical materials [141]. During electron beam evaporation, target material is hit and vaporized under vacuum by an intensive beam of electrons. A magnetic field is also applied to focus and change the beam trajectory. Large categories of coatings, such as amorphous and crystalline metals, oxides and molecular materials, can be produced [142]. Similarly, during laser beam evaporation a laser beam is used to ablate the material for depositing the coating and produce the plume with neutral or ionized species. The morphology of the film is affected by the substrate temperature, while the process is characterized by a fast deposition rate and compatibility with inert gases and oxygen for oxide film deposition. Several variations of laser beam evaporation such as pulse laser deposition [143] and pulse plasma deposition [144] for the deposition of ceramic films are also currently investigated. However, problems due to high temperatures and lack of uniformity occur. An electric arc at high current and low voltage that passes across the surface of the target is used to vaporize the material during cathodic arc deposition. The metal atoms become highly ionized, which makes the process suitable for the formation of thick films, but the released microparticles from the target can make the coating susceptible to corrosion processes.

Sputtering is an etching process involving the backward scattering of solid surface atoms upon bombardment by energetic ions to obtain glow discharge in the residual gas (reactive or inert) in the vacuum chamber. The sputtering process depends on the bombardment of the ions released from the discharge that hit the cathode to liberate particles (mostly neutrals) with high kinetic energy (Figure 8). The latter strike the substrate or anode to form a dense film [145]. Common sputtering systems adopt radiofrequency (RF), direct current (DC), magnetron diode, or ion beam sputtering. Although the deposition rate is lower than that of evaporation, the deposited films have a uniform dense structure, good adhesion to the substrate, high purity, outstanding adhesion to the substrate and composition similar to that of the starting material. Compounds such as oxides, nitrides and carbides can be formed by flowing reactive gases as well as composite materials such as Ag-HAp, Ag-SiC-HAp, Sr-HAp, Ti-Si-N, etc. [80]. However, it is hard to deposit uniform coatings on implants with complex shapes, as well as films with higher thicknesses and roughness. Even more importantly, the length of the process and its cost limit the sputtered coatings’ application.

The ion implantation technique is a process in which ions extracted from plasma by an extraction system accelerate in the form of a beam with high energy towards a substrate. Because of the small cross-sectional area of the beam, the beam must be rotated to achieve uniform implantation on flat large surfaces, whereas for complex geometries sample rotation is required which limits its size [141]. Plasma immersion ion implantation and deposition allows simultaneous and sequential ion implantation and deposition by combining conventional plasma and ion beam technologies. Under negative sample bias, positive ions are accelerated from the plasma while the plasma sheath around the sample governs the ion implantation process. In that way, advanced treatment of devices with complex shapes and large implantation areas for shorter process time is carried out [146]. This process was used to improve the biocompatibility and bioactivity of Ti6Al4V alloy by the formation of surface TiN and TiC layer [147], as well as to improve the corrosion resistance of magnesium alloys by adding metallic elements such as Zr [148] and metalloids like oxygen [149]. The method provides a strong adhesive bonding of the coating to the surface. Its main limitations are the formation of amorphous coatings and expensive costs [150].

Laser cladding is another electrophysical technique that uses a high-power density laser beam to melt the surface of the material which rapidly cools down to form cladding layers with changed surface structure and properties [151]. This process was used for the deposition of bioactive glass onto ultra-fine-grained Ti substrates [152], where the laser beam is focused onto the metallic substrate to melt it, while bioactive glass in powder form is delivered by inert gas. Due to the localized heating of the laser beam, the properties of the substrate are slightly changed but the homogenization of the microstructure in the cladded coating improves the mechanical properties and realizes metallurgical bonding with the substrate [153]. Although substantially roughened, the surface can exhibit pores and cracks.

Another electrophysical method is the electrospinning technique for the production of fibers with nano- or microscale diameters. Through a high-voltage supplier, a liquid droplet of a material solution is dispensed out of a nozzle of a syringe under a high voltage. At a critical voltage, the droplet becomes charged and stretches due to repulsion and surface tension [154]. The tip of the syringe act as an electrode while the collector is the counter electrode. Attracted by the counter electrode, the stretched droplet moves to the collector, while the solvent evaporates triggering the solidification of the material. Although polymers such as polycaprolactone (PCL) and polylactic-co-glycolic acid (PLGA) are typical starting materials for the production of fine fibers, metals, ceramics and composites can be also used [155]. The resulting filaments display the potential to serve as bioactive coatings due to the high porosity, the implementation of biocompatible and biodegradable materials and the incorporation of active ingredients. For example, loaded with antibiotics or other drugs, the coatings undergo initial burst release within the first few days and, later, slow controller release which can last more than a month [153,154,155,156,157,158].

### 5.5. Additive Manufacturing Methods

3D printing is a technology that constructs three-dimensional objects following a computer-aided design (CAD) or three-dimensional (3D) scanners in a layer-by-layer method [159]. The technique offers flexibility in terms of materials used, obtaining complex geometric shapes, time efficacy and low production costs [160]. Among various 3D printing technologies, the most commonly investigated methods for the development of bioactive coatings for orthopedic implants are extrusion-based methods such as fused deposition modeling (FDM) and semi-solid extrusion (SSE). During the FDM, the printed thermoplastic materials are softened and melted using a heated nozzle, built up layer-by-layer and, after solidifying, the filaments form a 3D coating on the metal surface. In SSE, semi-solids are, layer-by-layer, extruded using air pressure or screw gear rotation through a nozzle to form a 3D coating. A combination of FDM and hot melt extrusion is also used for the production of amorphous materials with active ingredients. Initially, the thermoplastic material and the active ingredients are mixed by pumping them at a temperature above the glass transition temperature with a rotating screw and after that FDM is printed to obtain the desired structures [161]. Additively manufactured composite coatings printed from cellulose nanofibril suspension, alginate and carboxymethylcellulose and further loaded with antibiotic clindamycin were deposited on SS 316 LVM and Ti6Al4V substrates [162]. Although not extensively explored for the production of bioactive coatings, 3D printing technologies are promising candidates for the development of personalized shelled systems for orthopedic applications or porous scaffolds for controlled drug release [163,164].

### 5.6. Combined Methods

The diverse applications of coatings have led to combining various conventional and unconventional assembly methods for the deposition of bioactive films. For example, in the layer-by-layer deposition method, the substrate is covered by oppositely charged constituents attached by electrostatic attraction, hydrophobic interactions, hydrogen bonds, or van-der-Waals forces by different methods [165]. They include the dip method of immersing in polymer or colloid solution followed by the spin method and spray method that allow faster layer deposition [166]. Recently, electromagnetic assembly applying electric current to electrodeposit the layers, fluidic assembly in which the substrate is fixed in the fluidic channels (tubes/capillaries) and the coating is deposited by the gradual movement of polyelectrolyte and rinse solutions by pressure or vacuum; as well as ink-jet printing, 3D bioprinting, inorganic-organic hybrid assembly, etc. have been developed [167]. In this way, layer-by-layer obtained coatings offer flexibility in production at mild temperatures, implementation of a wide range of materials and the ability for functionalization with peptides, drugs, etc. [168,169]. The number of layers and the incorporating method determines the kinetic profile of active substance release.

Table 1 summarizes some of the important advantages and disadvantages of the discussed deposition techniques for the deposition of bioactive coatings.

## 6. Bioactive Coatings

Surface modifications of metallic implants have been extensively explored to hinder a range of adverse effects such as long-term stability, low biomechanical properties, lack of biocompatibility, restriction of implant surface corrosion, post-surgery infections, etc. Therefore, the development and design of biometallics rely on surface modification since, by applying appropriate coatings, the surface properties can be tailored and improved. In connection with this, two approaches for surface modification have been applied: (a) direct coating deposition on the unmodified metal substrate; (b) initial substrate modification by grinding, sand-blasting, etching, or other treatment and deposition of overlaying coating. In the second case, improved surface roughness parameters for synergy of both textural properties and mechanical interlocking of coating are obtained.

The main requirements for a selection of coating material include (a) adequate mechanical reliability, adhesion strength and fracture toughness to withstand the applied forces; (b) corrosion resistance in the body fluid environment; (c) biocompatibility and lack of toxicity, allergic or other undesirable effects or inflammation [171]. Depending on their performance in the organism, the biomedical coatings can be subdivided into three main groups: bioinert, bioactive and bioresorbable [78]. In contrast to bioinert (such as Al_2_O_3_ and ZrO_2_), bioactive coatings refer to biomaterials that can stimulate the surrounding tissue and cells to regenerate around the exogenous graft and to release bioactive molecules for elimination the post-operative complications [172]. Absorbable (bioresorbable) coatings are designed to degrade in the human body via an electrochemical mechanism of dissolution and then metabolized by cells and tissue [173]. Recently published research works addressing osseointegration of inorganic bioactive coatings are given in the next paragraphs.

### 6.1. Inorganic Coatings

Various research groups focus on the development of inorganic coatings for biomedical applications because of their stability in the body environment, good mechanical properties, corrosion and wear resistance. To be called “bioactive”, these coatings should have surface-located functional groups that in an aqueous solution create conditions favoring heterogeneous mineral nucleation and growth on the surface. However, some of these inorganic coatings have shown disadvantages, including the release of toxic ions, cytotoxicity, lack of biodegradability, low bonding strength, etc. Some recent studies on the wide range of such inorganic coating materials are reviewed in the next sections.

#### 6.1.1. Nitrides

Various transitional metal nitride coatings such as TiN, ZrN, NbN, TaN, etc. have been studied as protective films against wear and corrosion of medical metal surfaces of the prosthesis and surgical implants [174]. Nitride films have a high melting point, chemical resistance to oxidation and acceptable adhesion [175]. Titanium nitride (TiN) films are often used in industry because of their high surface hardness and chemical properties. For biomedical applications, TiN was found to be well tolerated by tissue due to its inertness [176]. On orthopedic implants, nitride coatings protect the surface against wear and act as a diffusion barrier, preventing ion release from the metal to the body fluids [177]. Compared to other nitrides, TiN shows better biological properties for orthopedic applications [178]. However, because of dissimilarities in the hardness of TiN coating and substrate, plastic deformation at the coating/substrate interface occurs resulting in the formation of flakes, defects in the coating and fracture [179]. Compared with CVD-deposited TiN coatings, PVD utilizes a higher deposition rate and delivers improved bonding strength [180]. To improve the adhesion strength and wear performance “hard” nano-TiN and “soft” Ti_4_N_3−x_ transitional phase with variable composition was prepared by DC magnetron sputtering on Ti6Al4V alloy [181]. The coating showed excellent bonding and wear resistance because of the match of the mechanical properties at the substrate/coating interface together with good biocompatibility. Except for higher resistance to plastic deformation and improved wear behavior than the bare Ti20Nb13Zr (TNZ) alloy, TiN coating deposited by the cathodic arc PVD process indicated better corrosion resistance in SBF than TNZ [182]. Compared to bare Ti substrate, TiN coating showed approximately eight times more corrosion resistance and 4 times more wear resistance [183]. For that reason, sputtered TiN was used as an interior layer between HAp films and pure Ti rendering the controllability of thin film structural properties because of the improvement in bonding, strong fatigue resistance and better mechanical performance of the HAp layer [184].

Because the release of metal ions such as nickel, cobalt and chromium may cause serious problems in joint replacement due to metal hypersensitivity, considerable attention was also paid to PVD deposited ternary nitrides such as TiNbN that can act as a surface coat to hide the metal beneath affording an immuno-privileged state [185]. Copper-doped TiN (TiCuN) deposited by axial magnetic field enhanced arc ion plating has proved to own excellent corrosion resistance, wear resistance and good antibacterial properties and displayed no cytotoxic effect on human umbilical vein endothelial cells [186]. Compared with TiN coating, TiCuN promoted mRNA expression of endothelial nitric oxide synthase (eNOS) and vascular endothelial growth factor (VEGF), enhancing cell migration and angiogenesis ability. Similarly, ZrN/Cu coating deposited on SS and Ti substrate enhanced their wear- and corrosion resistance in SBF and their antibacterial activity by using ion release and a contact killing mechanism [187].

Similar to other transitional metals, Ta has a high affinity to nitrogen and forms inert TaN. TaN is also a commonly used material for the production of hard thin coatings because of its chemical inertness, corrosion resistance [188] and biocompatibility [189]. Mendizabal et al. discovered that the highest corrosion resistance was observed for Ta and low nitrogen content TaN_x_ films (lower than at30%) when deposited on AISI 316L steel by modulated pulse power magnetron sputtering [190]. The excessive amount of nitrogen on the film worsened its corrosion protection. Deposited on Ti substrate, the magnetron sputtered TaN film exhibited stronger bonding properties than TiN, and optimal compressive performance [191]. Deposited on Mg alloy by reactive magnetron sputtering, TaN exhibited a 95-fold decrease in corrosion current density in SBF solution compared to uncoated Mg-Y-Re alloy [192]. Additionally, the incorporation of Ag and Cu in TaN nanocomposite films gave the condensates anti-bacterial and anti-wear properties [193].

In contrast to crystalline TiN and TaN, amorphous SiN was also shown to be biocompatible and slow dissolving in a water-based solution [194,195]. It also has high hardness (up to 26 GPa) and Young modulus (up to 212 GPa), low wear rates, acts as a barrier for ion release from the metal and generates biocompatible wear particles [196]. However, some challenges such as inadequate adhesion and high dissolution rate that can reduce scratch resistance and cause premature coating failure are still faced by these coatings [197]. By alloying SiN with Fe and C in the Si-Fe-C-N system [198] or with Nb and Cr in the Si-Nb-Cr-N system [199], better adhesion, optimized dissolution rate and ion release can be obtained without compromising the adherence and morphology of MC3T3 or L929 cells. Overall, the composite coatings have low surface roughness, high hardness and elastic modulus and no evident cytotoxicity as opposed to SiN controls and CoCrMo alloy. With the increase of Cr between 4 to 6 at% the release of Si, Cr and Nb ions and the dissolution rate of the coating reduced, while the cell viability was reduced.

#### 6.1.2. Oxides

TiO_2_ is an important material in biomedical applications, since it has good mechanical properties, antibacterial and catalytic activity and long-term stability under photo- and chemical corrosion [200]. TiO_2_ can promote the formation of bone-like apatite or calcium phosphate on its surface when soaked in SBF solution which makes it suitable for bone reconstruction and replacement [201]. Moreover, it was found that the formation of TiO_2_ coating by anodization on the surface of Ti substrates was an effective method to reduce the temperature rise of the implant during microwave diathermy treatment that would provide a potential rehabilitation solution to internal fixation of bone fracture [202]. Similar to TiO_2_, tantalum oxide (Ta_2_O_5_) could facilitate the formation of bone-like apatite and stimulate the rapid attachment of bone and soft tissue [189]. Tantalum oxide produced by reactive magnetron sputtering was able to enhance both early-stage corrosion resistance and in vitro biocompatibility of Mg alloy [203]. Ta_2_O_5_ coating with 12.5 at% Ag deposited by a twin-gun magnetron sputtering system exhibited both improved antibacterial effects against *S. aureus* and good skin fibroblast cell cellular biocompatibility [204].

Grafting metal ions and compounds is also a common method to improve the osteogenic ability of oxide coatings, since various metal ions (Ca^2+^, Sr^2+^, Mg^2+^, etc.) have been demonstrated to possess the property of enhancing osseointegration [205]. An illustration of the influence of different metallic ions on various processes involved in bone regeneration is presented in Figure 9. Zhao and co-authors used MAO to produce Mn-TiO_2_ coatings on the Ti surface that showed good biocompatibility and osteogenic properties while the Mn^2+^ release from the coating promoted surface biomineralization [206]. TiO_2_ nanotubes produced by anodizing and loaded with Sr combined with icariin (ICA) showed a better effect on cell adhesion, proliferation and higher mineralization activity than pure Ti and TiO_2_ coatings. Furthermore, in osteoporotic rats, more bone was formed around the implants loaded with Sr and ICA [207]. Similarly, Y-doped TiO_2_ coatings on Ti6Al4V produced by PEO demonstrated good biocompatibility on osteoblastic precursor cells and fibroblast cells with increasing doping concentration of yttrium and excellent antibacterial activity against *E. coli* and *S. aureus* [208]. MAO processed Fe_2_O_3_/TiO_2_ composite coating on Ti implants with sensitivity to the micro-magnetic field because of super-para-magnetism, which was able to enhance fibroblast response including proliferation, phenotype and extracellular collagen secretion by increasing the amount of Fe_2_O_3_ NPs [209]. Compared to pure TiO_2_ coatings, Fe_2_O_3_ (4.41 wt% Fe)/TiO_2_ composite coatings reduced bacterial growth by 60% and efficiently prevented recession and inflammatory reaction of soft tissue. By immobilizing anionic polypeptides on the surface of TiO_2_ nanospike coating by coordination, Gao et al. demonstrated that the obtained film was able to kill pathogenic bacteria, inhibit biofilm formation for up to 2 weeks and promote the formation of HA on the surface [210].

TiO_2_ can not only be used to promote osseointegration but, also as a photosensitizer. As a stable photocatalyst, TiO_2_ produces ROS to kill bacteria under UV radiation but UV light is harmful to the body. For that reason, Nagay et al. prepared N- and Bi-codoped TiO_2_ coating on Ti by PEO that produced ROS to kill microorganisms under visible light [211]. By embedding silver (Ag) and zinc (Zn) nanoparticles into a 3D printed porous titanium oxide layer, the surface promoted the release of Ag^+^ and Zn^2+^ which favored antibacterial effect and osteogenesis, respectively [212]. Moreover, this synergetic effect was able to reduce the toxicity of Ag to the host cell. In contrast to bare Ti substrate and undoped TiO_2_, Ag-doped TiO_2_ coatings produced by sol-gel technology enhanced the corrosion resistance of Ti in SBF solution [213]. Nanoclusters of Ag incorporated in silica coatings obtained by RF co-sputtering technique displayed both good adhesion on steel substrate and antibacterial activity against *S. aureus* [214].

Aiming at modifying the surface of TiO_2_ to generate reactive oxygen species (ROS) to eradicate bacteria under near-infrared (NIR) light, different photosensitizers can be used. Chai et al. synthesized hydrothermally produced MoSe_2_ nanosheets on the surface of porous MAO-prepared TiO_2_ coatings and covered them with chitosan by electrostatic bonding to improve biocompatibility [215]. Under NIR irradiation because of the synergistic effect of hyperthermia and ROS generation, the coatings demonstrated excellent in vivo and in vitro antibacterial properties against *S. mutans,* whereas chitosan improved hydrophilicity and biocompatibility of the hybrid coating, promoting osseointegration even in the presence of infection under NIR light. Han and co-authors chose MoS_2_ with a broad spectral response to modify the surface of composite collagen/polydopamine/TiO_2_ coatings on Ti implants produced by MAO and hydrothermal treatment [216]. Under the combined action of photothermal and photodynamic therapy, the biofilm of *S. aureus* was quickly eradicated while the collagen promoted the adhesion and proliferation of osteoblasts. TiO_2_ nano-shovel/quercetin/L-arginine coatings doped with ytterbium (Yb) and erbium (Er) exhibited the production of ROS under near-infrared II light irradiation that could kill bacteria. At the same time, ROS catalyzed the release of nitrogen oxide (NO) free radicals from L-arginine which promoted angiogenesis and osseointegration [217]. The electrons and hole complexes generated by TiO_2_ reduced the photocatalytic properties, whereas the nano-shovel structure and quercetin that was coupled to the surface by organo-silanes, promoted the differentiation of bone marrow stem cells (BMSCs). Li et al. produced thermosensitive chitosan-glycerin-hydroxypropyl methylcellulose hydrogel (CGHH) to layer the top of simvastatin-loaded TiO_2_ nanotubes [218]. At 37 °C, the CGHH was found to be in a sol state which facilitated the controlled release of simvastatin to enhance MC3T3-E1 cell differentiation. The results of subcutaneous infection animal models indicated that CGHH had almost no antibacterial activity but, at high temperatures caused by infection, GCHH transitioned into a gel state and released a large amount of glycerin that induced acute inflammatory reaction and antibacterial activity against *S. aureus* and *E. coli*.

TiO_2_ can be used as a sound sensitizer to produce ROS by ultrasound-triggered electron-hole separation. Applying photoacoustic therapy sulfur-doped titanium oxide (S-TiO_2-x_) to titanium implants endowed the implant with good sonodynamic and photothermal properties [219]. Under NIR irradiation and ultrasound, the killing rate of *S. aureus* was equal to 99.995% after 15 min of exposure while the coating displayed good stability after soaking in water for 6 months.

The principle weakness of bio-ceramics originates from the low mechanical strength that makes them inappropriate for load-bearing application. When combined with metallic implants and bio-ceramic films, their mechanical properties are preserved while the integration with the bone is improved. However, the metal–ceramic interface accumulates residual stresses causing delamination at the interface.

#### 6.1.3. Oxynitrides

To combine superior mechanical properties such as hardness and adhesion to the substrate and enhanced corrosion resistance, oxynitrides of transitional metals have been developed. Moreover, transitional metal nitride-oxide coatings are interesting materials because of their low degree of dissolution, corrosion resistance and inertness in body fluids [220]. The oxygen addition in cathodic arc deposited TiN decreased the grain size and enhanced the formation of a passive layer resulting in superior corrosion resistance in aggressive H_2_O_2_ augmented saline solution [221]. Sputtered TiN_x_O_y_ coatings with chemical composition ranging from TiN to TiO_2_ deposited on microroughened titanium plates showed a significantly high level of bioactivity as compared to bare Ti substrates (1.2 up to 1.4 fold increase in cell proliferation) that made them biocompatible over a broad range of compositions [222]. Similar results were reported for TiN_x_O_y_ coatings deposited on roughened SS [223] and CoCr alloy during the first two weeks of healing [224]. Therefore, in addition to the enhanced wear resistance of TiN_x_O_y_ coatings, they can “isolate” the substrate metal from the bone and accelerate the effect on the growth of bone cells. For that reason, our research group synthesized gradient TiN/TiO_2_ coatings by a cathodic arc (for TiN) and glow discharge deposition (for TiO_2_) on an initial surface treated by electron beam Ti6Al4V alloy [225]. The initial electron beam treatment (EBT) of the alloy not only roughened the surface of the alloy forming regular grooves and heights but also enhanced the hardness of the substrate, thus generating a smooth gradient in stiffness from the substrate to the coating. Because of the decreased grain sizes and increased number of defects in the substrate, the EBT lowers the heat conductivity (λ) of the surface. Therefore, due to trapping heat near the surface, this initial substrate treatment not only improved the adhesion of TiN/TiO_2_ coating to the Ti6Al4V alloy but also triggered reorientation in the micro-volumes of the nitride and rutile to anatase ratio of the oxide (Figure 10), thus decreasing its surface hardness and bringing it closer to that of trabecular bone and human teeth and decreasing the elastic modulus mismatch between the bone and implant.

We observed similar reorientation of the micro volumes of the nitride and oxide and a decrease in microhardness after EBT for the magnetron sputtered TiN/TiO_2_ coating deposited on both Ti5Al4V alloy [226] and Co-Cr alloy [227]. The TiN/TiO_2_ coatings displayed adhesion bonding between the nitride and oxide layers [228] with substantially improved tribological performance and corrosion resistance as opposed to the Ti-Al-V substrate. Compared to the magnetron sputtered TiN/TiO_2_ coatings, those deposited by cathodic arc deposition (CAD) and glow discharge oxidation showed better human osteoblast-like cell (MG63) adhesion, viability and bone mineralization activity on both polished and EBT Ti6Al4V samples (Figure 11). This is because, in contrast to the less rough magnetron sputtered (MS) TiN/TiO_2_ coatings, on the surface of CAD and glow discharge oxidized coatings there are many surface elements for focal adhesion in the coating, such as droplet phase particles, disoriented crystallographic planes at the tips of oxide crystals, defects such as pores, etc., all of which support cell movement and proliferation. Simultaneously, cells cultured on the grooved surface with a smaller channel spacing (AR500) tended to have a stronger orientation along the groove axis compared to the AR850 surface with greater groove spacing. The results show that the deposited TiN/TiO_2_ coating on the micro-rough EBM surface stimulates and accelerates cell differentiation [225].

Besides, magnetron-sputtered TiON and ZrON films on SS 316L substrates indicated a drastic reduction of bacterial adhesion (P. aeruginosa) as well as inhibition of biofilm formation at different time durations [229]. The bactericide activity of TiON and TiON-Ag sputtered films under visible light irradiation was reported by Rtimi et al. [230]. They stated that compared to TiON film which inactivated bacteria within 2 h, TiON-Ag coatings with Ag concentration below the cytotoxicity level showed faster and repetitive inactivation of *E. coli*. Similarly, ZrO_2_-Ag and ZrON-Ag coatings had lower bacterial retention while ZrON-Ag with porous structure and 11.8 at% Ag possessed the best antibacterial performance against *S. aureus* and *A. actinomycetemcomitans* together with excellent human gingival fibroblast (HGF) cell compatibility [231]. For magnetron-sputtered ZrON-Cu coatings, only the presence of CuO species caused bactericidal activity against S. epidermidis while Cu^2+^ ion release did not influence the antibacterial properties of the coating [232].

#### 6.1.4. Carbon-Based Coatings

There are several types of carbon-based materials that are used for biomedical applications: (a) amorphous carbon nanostructures (diamond-like carbon (DLC), graphite-like carbon (GLC), pyrolytic carbon); (b) nanocrystalline diamond (NCD); (c) graphene and its derivates. Recent studies indicate that these carbon materials have exceptional biocompatibility, stability and mechanical properties [233,234]. Amorphous carbon with high sp^3^ content is referred to as diamond-like carbon (DLC) while higher sp^2^ content yields materials closer to graphite (graphite-like carbon). However, in contrast to graphite, the latter has higher hardness and high corrosion resistance. Pyrolytic carbon is also an amorphous carbon allotrope with dominating sp^2^ bonding. It is conventionally produced by CVD from gaseous hydrocarbon precursors. Depending on the process conditions, pyrolytic carbon coatings can have isotropic, granular, lamellar, columnar, etc., structures. Although mainly used for heart valve protection [235], pyrolytic coatings have been applied for the replacement of small joints such as knuckles, wrist joints and proximal interphalangeal joints [236].

DLC-based coatings are considered promising for bioimplant application because they have excellent mechanical properties, a low coefficient of friction and good wear resistance. For that reason, applied as coatings on Ti substrates by a CVD technique, the DLC film substantially improved the nano-hardness and tribological performance, decreasing the coefficient of friction by one order of magnitude [237,238]. The in vivo behavior of PVD-deposited DLC coatings on Ti substrates indicated no inflammatory reactions, confirming its good biocompatibility [239]. However, some disadvantages such as high internal stress, low toughness and high sensitivity to ambient conditions can be observed for a single layer of DLC coating that can explain the high revision rates of single-layered DLC-coated orthopedic joints [240]. Compared to TiN-coated joint prosthesis, DLC coatings demonstrated lower wear resistance [241]. To address this problem, multilayered coatings on Ti-6Al-4V alloy consisting of (a) alternating Zr and ZrN sublayers responsible for corrosion resistance and load carrying capacity, (b) overlaying Zr/DLC composite film for enhanced adhesion and reduced fatigue residual stresses and (c) top N-doped DLC to reduce friction and enhance, have been designed [242]. The resultant coatings showed a decreased coefficient of friction by more than 50% and two to three times increased hardness than that of bare Ti substrate. Except for a substantial decrease in wear, the middle layer improved the delamination strength, which is low in single DLC coatings. Additionally, fluorinated DLC coatings also exhibited good antibacterial properties against *E. coli* and *S. aureus* by decreasing their counts from 2.4 *×* 10^4^ and 2.54 × 10^4^ to less than 20, in contrast to two orders of magnitude growth of bacteria in the control groups [243]. At the same time, no substantial difference in cytotoxicity between the groups was observed confirming good biocompatibility of the coating. To improve the biocompatibility of magnetron-sputtered DLC coatings, Si-doping was also applied. Deposited on Ti6Al7Nb alloy, the addition of silicon up to 14–22 at% to the DLC coatings had a very positive effect on the proliferation and viability of endothelial cells [244]. Increasing the Si content resulted in a rise in the hydrophilic character of the coating, film hardness by up to 40% and reduced colonization by *E. coli* bacteria compared to the uncoated substrate [245]. Wachesk et al. deposited hybrid DLC coatings incorporating TiO_2_ nanoparticles by plasma-enhanced CVD on AISI 316 and implanted them in CF1 mice peritoneum [246]. The in vivo results showed that the presence of TiO_2_ nanoparticles enhanced healing activity and reduced the inflammatory reactions increasing DLC biocompatibility. However, a major concern with DLC coatings is their instability in an aqueous environment, which promotes delamination of the coatings [247].

Nanodiamonds possess a high surface-area-to-volume ratio together with good biocompatibility and bioactivity [248]. Additionally, diamonds were reported to have high wear resistance and low friction coefficient, which make them ideal for protective layers [249]. Nanodiamond coatings also show high surface roughness, hydrophobicity of the surface, high stability, superior electrochemical properties and biocompatibility [250,251]. On metallic substrates nanocrystalline diamond (NCD) coatings behaved as well-adhering and highly cohesive films [252]. It was found that the cell performance on NCD films depended on surface atoms or chemical groups [253]. For example, on micropatterned NCD films, human dental stem cells adhered and grew preferentially on O-terminated domains rather than on H-terminated areas [254]. The low boron (100 to 1000 ppm of B) doping of NCD films was also found to support cell proliferation and early osteogenic differentiation of MG63 cells because of the increased electroconductivity of the doped films [255]. A similar effect of enhanced attachment and spreading of MG63 cells was observed for composite apatite-nanodiamond coatings compared to pure SS and apatite coatings without nanodiamonds [256]. The authors explained the observed effect of the increased adsorption of fibronectin on the composite coatings. Simultaneously, Medina and co-authors observed that NCD coatings were able to establish a chemical bond with the cell wall or membrane of Gram-negative P. aeruginosa bacteria, thus hindering the bacterial adhesion and colonization of the surface [257].

Two-dimensional (2D) allotropes of carbon–graphene, a single atom thick layer of sp^2^ carbon and related graphene oxide (GO) and reduced graphene oxide (rGO) are innovative materials in the medical sector because of their unique biological properties. Graphene oxide is an oxygenated derivate of graphene with abundant functional groups on planes and edges allowing desirable dispersion behavior in aqueous media [107]. rGO consists of fewer oxygen-containing groups because of interactions with reducing agents [258]. The chemical properties of rGO resemble those of pristine graphene [259]. Many scientific works report the ability of rGO to promote osteogenic stem cell differentiation [260,261]. Graphene can be directly grown on metallic surfaces such as Ti6Al4V [262] and Mg [263] to improve bioactivity and corrosion resistance. It is also popular material with antibacterial, antifouling and hemo-compatible properties but the layered structure of graphene nanosheets limits its benefits and advantages [264]. For that reason, graphene nanoplatelets with improved biocompatibility and effectiveness for biomedical devices have been introduced [265]. In such a form, graphene is usually combined with natural or synthetic biopolymers to enhance the osteogenic potential and mechanical properties of the coating. For example, by using electrophoretic deposition, Suo et al. deposited GO/chitosan/HAp coatings on Ti that showed higher bonding strength to the substrate than HAp, GO/HAp and chitosan/HAp coatings and significantly enhanced cell–coating interactions in vitro and osseointegration in vivo [261,262,263,264,265,266]. Simultaneously, the fracture toughness of HAp rose by 200% by including only 1 wt% rGO [267]. Graphene-based materials have powerful antimicrobial properties and inhibit bacterial colonization. For example, Agarwalla and co-authors [268] tested graphene coatings on Ti against *P. aerugimosa*, *E. faecalis*, *S. mutans* and *C. albicans* and found that, when repeated twice, the film reduced biofilm formation due to the hydrophobicity of graphene. Similarly, functionalized GO nanocomposite with Ag NPs showed excellent antimicrobial activity against *E. coli* and *S. aureus* [269]. Despite these impressive properties, there is still a concern about the biodegradability of nanodiamonds and graphene in the organism. Additionally, in vitro studies with GO nanomaterials indicated the generation of ROS, DNA damage and mitochondrial disturbance [270].

#### 6.1.5. Calcium Phosphates and Hydroxyapatite

Calcium phosphate ceramic coatings are extensively used to boost the biocompatibility of metal implants because of their superior adaptation to in vivo conditions. Bioactivity properties are varied according to the type of calcium phosphates. Both calcium phosphate types, HAp and tricalcium phosphate (TCP) have different crystallinity, stability, solubility, ion release and mechanical properties. The crystallinity is affected by the Ca/P ratio and a higher amount of Ca^2+^ or PO_4_^3-^ can trigger amorphous phase formation such as dicalcium phosphate dihydrate, CaHPO_4_.H_2_O and Ca_3_(PO_4_)_2_ [271]. In contrast to highly crystalline HAp, calcium phosphate-based coatings have high solubility in an aqueous medium that reduces coating stability and can cause implant loosening. Because of its similar properties to the inorganic composition of hard tissue such as bone and teeth [272], HAp (Ca_10_(PO_4_)_6_(OH)_2_, Ca/P = 1.67) is frequently employed as bioactive material. HAp has shown exceptional biocompatibility, osteo-inductivity, osteoconductive and bioactivity [273]. By releasing calcium and phosphate ions, HAp enhances bone regeneration and promotes mineralization [274]. By covering the metal biomaterial, HAp coating helps in maintaining stability fixed to the bone while minimizing the side effects of ion release in the bio-environment. The enhanced osteoconductive properties of HAp coatings can be attributed to the bone-like apatite chemistry of the coating (Na^+^, Mg^2+^, CO_3_^2−^, Ca^2+^ and PO_4_^3−^) and reduced degradation rate that allows a balance between ion release by the coating and ion absorption by the tissue during the bone formation [275].

However, due to their ceramic nature, highly porous or highly crystalline HAp coatings can show low mechanical properties (very brittle, with low flexibility), poor adhesion to the metal surfaces and low corrosion resistance, which makes them inappropriate for load-bearing applications. The difference in the thermal expansion coefficient of metallic alloys and HAp results in residual thermal stresses which can promote cracking or delamination of the coatings. The corrosion resistance also depends on the deposition method. Sankar et al. compared the corrosion behavior of HAp coatings obtained by EPD and pulse laser deposition (PLD) method and found that EPD coating had lower corrosion protection than PLD films due to the formation of denser and pore-free structures [276]. Dispersion strengthening by introducing a second phase to its microstructure such as other ceramics, carbon nanotubes or other compounds is deemed to overcome the poor mechanical properties of these coatings. For example, the addition of TiO_2_ to HAp (20–80 wt%) coating produced by High-Velocity Oxygen Fuel (HVOF) spraying on Ti6Al4V alloy delayed the HAp dissolution and increased the coating stability in Hank’s solution [277]. Similarly, by introducing TiO_2_ to fiber HAp by the EPD process it was discovered that the pores of the HAp coating produced from suspension with 50 and 75 wt% fiber HAp can be efficiently infiltrated and filled with TiO_2_ nanoparticles which increase the corrosion resistance of the coatings in SBF solution [278]. Evcin et al. [279] produced a series of HAp/Al_2_O_3_, HAp/B_2_O_3_ coatings on Ti6Al4V alloys by the HVOF method and found that increasing the amount of Al_2_O_3_ and B_2_O_3_ in HAp increased the adhesion strength and wettability.

The addition of specific trace elements to HAp like Zn, Mg, Cu, Si, Sr, Mn and F can also have a role in bone regeneration. For example, Zn was found to increase alkaline phosphatase activity and stimulate bone formation by osteoblasts [280]. For that reason, Zhou and co-authors produced Zn-doped HAp coating on ZK60 magnesium alloy by one-pot hydrothermal method with nano-whisker structure and showed that the film had a higher corrosion resistance compared to HAp coatings, promoting adhesion and differentiation of rat bone marrow mesenchymal stem cells at Zn concentration of 5% and obvious antibacterial activity [281]. Si-doped CaP was deposited on AZ31 magnesium alloy and the osteoblast cytocompatibility, evaluation showed that Si ion played a vital role in the nucleation and growth of apatite thus influencing the biological metabolism of osteoblast cells [282]. Pure Mg demonstrated an antimicrobial effect because of the increase in pH by degradation, while F is a basic element in bones. F can stimulate the differentiation of mesenchymal stem cells into osteoblasts, induce bone formation and promote the nucleation of HAp [283]. Comparing F-doped HAp, Mg-doped HAp and Mg/F-doped HAp coatings deposited on Ti substrate by pulsed laser deposition, it was found that Mg-F-HAp coating better promoted the transformation of apatite-like to HAp phase due to the synergistic effect of Mg and F. The porous 3D structure of the coatings enhanced the viability of rBMSCs, especially for Mg-F-HAp coatings, where a regulated biodegradable rate and good cellular proliferation were observed [284]. Sr ions were also found to increase bone-to-implant contact by osteoblastic cell proliferation, accelerate bone matrix synthesis and inhibit bone resorption [285]. Consequently, Sr-doped HAp coatings prepared on Mg-4Zn substrates by electrochemical deposition showed better corrosion resistance, improved protein adsorption and initial adhesion of mesenchymal stem cells, as well as improved osteogenic differentiation compared to HAp coatings [286]. Similarly, Ca-Sr-P coatings with dense crystalline structure deposited on biodegradable Mg alloy by chemical immersion method demonstrated improved corrosion resistance, higher bone formation and better osteointegration around the coating than the Mg alloy after 4 weeks of implantation in a rabbit model [287]. The biocompatible properties of HAp coatings were also enhanced by imparting antibacterial properties by incorporating silver. For example, in the presence of F in Ag-F-HAp coatings developed on Ti substrate by sol-gel method with a silver concentration of 0.3 wt%, the crystalline size and pores in the coating decreased whereas the antibacterial activity against *E. coli* bacteria increased with the amount of F [288].

The incorporation of polymers in HAp coating structures was also found to have a positive effect on the ceramic coating properties. For example, electro-phoretically deposited HAp-CaSiO_3_-chitosan composite coatings that were made porous by heat treatment at 700 °C in a controlled atmosphere indicated improved corrosion resistance and bioactivity in SBF compared to Ti substrate [289]. Similarly, biomimetically deposited Ce-doped HAp/collagen coatings on initially alkali-thermal oxidation pretreated Ti substrate showed good antibacterial activity against both *E. coli* and *S. aureus,* being more effective against *E. coli* [290]. By electrostatic interaction, the negatively charged surface absorbed positively charged collagen and negatively charge HAp that incorporated Ce ions in its lattice.

The general requirements for the properties of HAp coatings are listed in Table 2, although the relatively high thickness of HAp coatings, the ease of delamination of the film from the base metal and the possibility of the coating fracture occurrence can reduce the functional performance of the implant. The debris may cause inflammation in the host body.

#### 6.1.6. Bioactive Glasses

The bioactive ceramic materials include silica, calcium, phosphorous and sodium ions (glass composition CaO-SiO_2_-P_2_O_5_-Na_2_O) that are released when the bioactive glass interacts with cells and the bio-environment, leading to fast bone growth. The 45S5 bio-glass (45 wt% SiO_2_, 24.5 wt% CaO, 24.5 wt% Na_2_O, 6 wt% P_2_O_5_) has shown the most effective bioactive properties, namely class A bioactivity, allowing it to bond to soft and hard tissues [292]. Similar to many other ceramic materials, bioactive glasses can be produced as particles with micron and nano-size, or fibers, 3D scaffolds, mesoporous coatings, or monoliths [293]. Compared to other biomaterials, bioactive glasses can make possible better integration between the metal implant and the growing tissue because of their significant bioactive behavior [276]. For example, when comparing the in vivo efficacy of CaO-MgO-SiO_2_-based bioactive glass-ceramic on Ti6Al4V alloy (deposited by atmospheric plasma spraying) with HAp-coated samples implanted in New Zealand rabbits, the significant growth of new bone confirmed the superior biological activity of bio-glass coatings in treating load-bearing bone defects [294].

Although showing excellent bioactivity, because of their semicrystalline or amorphous structure, bio-glasses can exhibit poor mechanical strength, low tensile strength, fatigue resistance, elastic modulus and corrosion resistance especially as regards porous coatings. The porosity formation of coatings obtained by plasma spraying occurred due to the evaporation of volatile Na_2_O and P_2_O_5_ [295]. To overcome this disadvantage, composites with metal oxides such as ZrO_2_, TiO_2_, Al_2_O_3_ or graphene and its derivates were made [296]. These could improve the thermal, electrical and strength properties of bioactive glasses [297]. Additionally, except for excellent bioactivity, such bioactive glass composite coatings exhibit improved antibacterial activity, angiogenic properties and corrosion resistance [298]. For example, one-dimensional bioactive glass nanorods of 45S5 composition produced by sol-gel process and hybridized with reduced graphene oxide sheets (rGO), following different methods for developing composites such as constant stirring, sonification and simultaneous reduction in GO–bio-glass composite, showed better results in bioactivity, hemocompatibility, cell proliferation and antibacterial activity as compared to pure bioactive glass nanorods [299]. Similarly, electro-phoretically deposited bioactive glass-rGO hybrid thin films (2 μm thickness) deposited on TiO_2_ nanotubes with a diameter of around 100 nm were advantageous in antibacterial activity, hemocompatibility and MG-63 cell proliferation [300].

Bioglass composites with additions of metals, metal oxides and HAp also demonstrated promising bioactive properties. For example, laser process bio-glass coatings reinforced with Ti on Ti substrate with excellent coating interfacial characteristics, improved hardness, corrosion protection and in vitro wear resistance, also indicating better cell-material interaction than bare -Ti [301]. However, when comparing HAp-based HAp/Ag coatings with bio-glass-Ag bio-composite coatings on NiTi alloy, both deposited by sol-gel method, higher corrosion resistance and adhesion strength were found for HAp/Ag coatings [302]. Often, there are challenges in creating a good adhesion between the glass topcoat and the metal substrate due to low metal–amorphous ceramic interface bonding and the formation of cracks [303]. A solution to the problem can be the utilization of a polymeric matrix to create a nano-composite coating. These are characterized by low processing temperature and elimination of the sintering process if required. Among various polymers, chitosan, a natural polymer, is often used because of its biodegradability, biocompatibility, osteo=conductivity and antimicrobial properties [304]. Recently, Alaei and co-authors produced chitosan-bioactive glass nanocomposite coatings that provided significant corrosion protection to Mg alloy and controlled biological properties [305]. A new family of chitosan-based composite coatings incorporating HAp–bio-glass and different concentrations of Fe_2_O_3_ particles was electro-phoretically deposited on Ti13Nb13Zr alloy [306]. All Fe_2_O_3_-containing coating formulations showed favorable hemocompatibility, better surface properties, improved corrosion resistance and better cytocompatibility with MG63 cells as opposed to bare alloy and Hap–bio-glass coatings. Similarly, Mn-modified bio-glass/alginate nanostructure composites deposited on SS 316L by electrophoretic deposition demonstrated that the increased manganese in bio-glass had a positive effect on corrosion resistance in SBF and improved bioactivity [307].

### 6.2. Organic Coatings

Recently, the interest in applying polymer materials as coatings has increased substantially because of their easy fabrication, affordable price, low toxicity, corrosion resistance and eco-friendly nature. Polymers show low strength and elastic moduli as compared to ceramics and metals and are not used for load-bearing applications. They can be both non-biodegradable and biodegradable with complete degradation over time. However, except for low mechanical properties, another issue faced by polymers is their inadequate degradation rate and inflammatory reaction which limitations prevent them from being widely used as biomaterials for hard tissue coatings [57].

#### 6.2.1. Synthetic Polymers

The most commonly used synthetic polymers for periosteum development are polylactic L-lactic-co-glycolic acid (PLGA), polyurethanes (PU), polyethylene glycol, (PEG), polycaprolactone (PCL) and poly L-lactic acid (PLLA), and polymethylmethacrylate (PMMA) [308]. Synthetic polymers are usually hydrophobic and possess no antibacterial activity. They can also deteriorate the adhesion of bone cells [309] restricting their widespread application in the medical sector. To improve the biological performance of polymer coatings, composite systems based on biocompatible polymers modified with various compounds or particles are often used [310]. For example, combining PU, which is usually used in medicine because of its favorable mechanical properties and high biocompatibility, with 0.25 wt% graphene (used as an antibacterial agent) and 2 wt% β-TCP (as a bioactive component) in dip coatings on Ti implants gave positive cell response in normal human osteoblast (NHOst) cells and effective antibacterial activity in contrast to the other examined composites with higher graphene content [309].

Similarly, PMMA is characterized by high thermal and chemical stability, biocompatibility and advanced mechanical properties [311]. Extensive studies on PMMA composites containing HAp, metal oxides and bio-glass showed that they are attractive for surface modification of biomedical implants because of their high biocompatibility, bioactivity and antimicrobial properties [312]. High molecular PMMA composite coatings with TiO_2_, Al_2_O_3_, HAp, bio-glass and Hap–bio-glass also provided enhanced corrosion protection compared to pure PMMA coatings [313].

Conductive polymer coatings have been also used as coatings on hard implants. Among them, poly-pyrrole (PPy) which has good biocompatibility, is frequently examined for biomedical applications [314]. However, once fabricated, pristine PPy has a rigid, brittle and insoluble nature [315]. To overcome this shortcoming, composites with various additives have been developed. For example, a composite coating of PPy with ZnO was developed by Guo et al. to protect the biodegradable Mg alloys from fast decomposition and to impart cyto-compatible and antibacterial properties [316]. Multifunctional composite coatings of PPy with pectin and 10 wt% gentamicin deposited on TiNbZr substrate demonstrated effective antibacterial performance, lower corrosion rate, controlled degradation because of the slow release of gentamicin and improved biocompatibility [317].

A synthetic polymer matrix was also used to augment HAp and bio-glass coatings and improve their mechanical strength. For example, nanostructured HAp was incorporated in polyetheretherketone (PEEK) to form PEEK–HAp composite coating which was deposited by EPD on 316 SS and heat treated at 375 °C to densify the coating and enhance the adhesion to the substrate [318]. In contrast to the as-deposited film where the HAp covered the PEEK and stimulated bioactivity, after heat treatment the HAp became encapsulated in PEEK and reduced bioactivity. Both adhesion strength and bioactivity were dependent on PEEK/HAp ratio. The increased amount of HAp caused improved bioactivity and reduced adhesion strength. Biodegradable PCL coatings on 316L SS containing 10 wt% gelatin (GE) and 3 wt% bio-glass showed drastically improved corrosion resistance and significant apatite formation as opposed to only PCL/GE coatings [319]. The bio-glass-containing composites also revealed increased MG63 cell viability compared to PCL/GE coatings while the results in an animal model (New Zealand white rabbits) demonstrated no inflammation and granulation, endothelial swelling, fibrotic tissue or other toxic effects.

More merits can be offered by biodegradable and resorbable polymers than non-degradable ones in terms of low levels of possible infections and implant rejection. However, the degradation of polyesters such as polylactic acid (PLA) and polyglycolic acid (PGA) and their co-polymers is known to create an acidic environment. The process can trigger host tissue response and foreign body reactions during degradation, as well as moderate cytotoxic reactions [320]. For that reason, cationic polymers with low toxicity such as poly(glycidyl methacrylate) or PGMA can be used. When Ti implants were functionalized with PGMA coupled with quaternized polyethyleneimine (bactericidal agent) and alendronate with high affinity to bone minerals, the obtained coating inhibited bacterial infections and promoted osseointegration in the late stages [321]. However, synthetic polymers do not have signaling sequences that are naturally present in biological polymers such as collagen, fibrinogen or fibronectin.

#### 6.2.2. Polymeric Gels (Natural Polymers)

Various natural polymers such as chitosan, silk fibroin, collagen, etc., have been used in the production of bioactive coatings. Chitosan, representing the de-acetylated derivative of chitin, is considered to have osteoconductive properties. Coatings of chitosan/heparinized GO deposited by layer-by-layer technique on Mg alloy showed that substantial endothelial cell adhesion and proliferation were promoted [322]. Chitosan-Mg composite dip-coatings on Mg-Gd alloys also showed a higher amount of newly formed bone in rabbits [323]. Similarly, improved cell adhesion and proliferation of osteoblasts were observed on coated AZ31D alloy with bioactive carboxymethyl chitosan by immersion treatment [324]. As cationic macromolecule, chitosan can bind to the negatively charged cell membrane of bacteria and display, albeit weak, antibacterial properties. To impart a stronger bactericidal effect of the chitosan-containing coatings, films of chitosan and hyaluronic acid (HA) on rough Ti substrate were designed to release β-amino acid-based peptidomimetic antimicrobial peptide [325]. The layer-by-layer prepared coating showed a strong chemical cross-linking of chitosan with HA films which caused prolonged β-peptide retention that selectively prevented *S. aureus* biofilm formation for up to 24 days and remained its bactericidal properties after being challenged sequentially five times with *S. aureus* inoculum over 18 days. Simultaneously, no significant cytotoxicity on osteoblast precursor cell line derived from mouse (MC3T3-E1) compared to uncoated and film-coated controls without β-peptide was observed. Such a novel localized delivery approach that can maintain long-term antibacterial properties is promising for the development of coated medical devices prone to biofilm-associated infections. However, the adhesion and durability of chitosan coatings might raise some concerns [326].

Polydopamine, the final oxidation product of dopamine or other catecholamines, was found to form layers with an adjustable thickness (from a few to about 100 nm [327]) with good adhesion and high cell affinity [328]. This fact was confirmed by the study of Peng and co-authors who demonstrated an enhanced osteogenic differentiation on Zn-containing polydopamine films on AZ31 magnesium alloy together with improved osteogenesis and osteointegration in Sprague-Dawley rats after 8 weeks post-implantation [329]. A hybrid coating consisting of hydrothermally grown ZnO nanorods on Ti modified with polydopamine and covalently immobilized Arg-Gly-Aspartic acid-Cys (RGDC) peptide promoted cytocompatibility, new bone tissue formation and osteointegration between the implant and the new bone even in the presence of injected bacteria, or demonstrated simultaneous osseointegration and infection prevention [330]. Polymer coating produced via reversible addition-fragmentation chain transfer polymerization from glutamic acid and dopamine metha-crylamide was immobilized on Ti substrate by catechol pendants on the polymer chain [331]. Besides promoting mineral deposition, the coating was found to promote osteoblast adhesion and proliferation. Dopamine-silver loaded coating prepared at different pH values (4, 7 and 10) and different Ag^+^ concentrations (0.01 and 0.1 mg/mL) showed that the pH10/0.1 group displayed osteogenesis in the bacterial environment due to the great antibacterial properties and promoted mineralization activity [332]. To reduce the well-known cytotoxicity of Ag, Guo et al. prepared Poly-L-lysin (PLL)/sodium alginate/PLL self-assembled coating loaded with nano-silver on Ti that effectively inhibited the adhesion of bacteria [333]. At the same time, the PLL/SA/PLL coating induced mineralization in SBF and improved cytocompatibility and reduced cytotoxicity. Similarly, by double chelation of dopamine and chitosan, a hybrid coating consisting of HAp/dopamine/chitosan and nano-silver achieved a long-term release of silver and a continuous bacteriostatic effect [334]. This effect was accompanied by substantial osteogenic potential demonstrated in both in vitro and in vivo tests.

Silk fibroin consists of light and heavy chains and hydrophobically linked glycoprotein P25 that are all crosslinked to form a complex with antiparallel beta-sheets [335]. Because of the formation of β-sheets, silk fibroin scaffolds have better mechanical properties than collagen and chitosan but they are still insufficient compared with bone tissue [329,330,331,332,333,334,335,336]. In the form of hybrid coatings on WE43 magnesium alloy consisting of an inner layer of Mg(OH)_2_ produced by anodization, a middle layer of HAp formed by EDP and an outer silk fibroin layer deposited by spin coating, the surface modification was shown to improve not only corrosion resistance but also cell attachment, viability and proliferation [337]. To increase the osteogenic capacity and mechanical properties of silk fibroin, besides different organic and inorganic components, surface modifications by bioactive moieties that form hybrid films can be applied. For example, blends of silk fibroin/chitosan/rGO were fabricated by solvent casting method as films whose hydrophilicity, swelling and degradability decreased with increasing silk fibroin content, whereas the tensile strength increased [338]. The cell behavior of the G-292 cell demonstrated promoted osteogenic performance by increasing chitosan content while the increase in rGO reduced the porosity and tensile strength. The optimum result corresponded to SF:CS:rGO equal to 84:7:9 weight ratio.

#### 6.2.3. ECM Proteins/Cell Coatings

Synthetic or natural multifunctional peptides can be used as coating materials on metallic grafts because introducing organic molecules that contain functional fragments can stimulate the interaction with proteins of the extracellular matrix (ECM). The organic part of ECM consists of collagen type I fibrils embedded in the amorphous substance of glycosaminoglycans and different bone proteins. Since ECM components actively participate in the regulation of cellular processes and interactions, the modification of the implant surface with components of the ECM is an attractive approach. Collagen is known to enhance tissue regeneration of bone, tendon, ligaments and vascular and connective tissue [339]. Collagen type I coating extracted from rat tail and deposited on Mg-Zr-Ca alloy implants by dip-coating showed accelerated protein bonding capacity resulting in better osteoblast activity and a tendency to form superior trabecular bone structure in male New Zealand white rabbits compared to the uncoated samples in a shorter period of implantation [340]. Another derivate of extracellular matrix proteins promoting cell adhesion as integrin ligand [341] is RGD. By using high-affinity inorganic peptides such as TiBP that contain a Ti-binding domain, RGD can be combined with antibacterial peptides to form durable and stable coatings with both bone-promoting and antibacterial properties [342]. Similarly, Zhang et al. used TiBP to connect antimicrobial sequence from human β-defebsin-3 and RGD in a coating that was found to significantly reduce the bacterial colonization onto the Ti surface and better-supported MC3T3-E1 cell growth compared with PBS-treated Ti samples [343].

Not only proteins but also vesicles and cells can be immobilized on the surface of metallic implants to form biogenic coatings. For example, using secreted extracellular vesicles (EVs) by mesenchymal stem cells, it was found that tissue repair and regeneration can be promoted since their membranes contained signaling molecules and, additionally, EVs can carry and transfer different cargos [5]. For that reason, Chen et al. immobilized adipose-derived stem cell extracellular vesicles with physisorbed fibronectin onto the Ti surface and observed enhanced osteoblast compatibility and osteo-induction activity [344]. Another approach included the immobilization of Lactobacillus casei on the surface of heat-treated Ti to form a probiotic coating [345]. The polysaccharides in the film promoted osteogenic differentiation through immunoregulation of macrophages that secreted osteogenic factors, while the surface showed 99.98% antibacterial effectiveness against *S. aureus*. However, the limitations of all these organic coatings are related to low mechanical strength, difficult sterilization and rapid degradation that can be overcome by designing composites with bio-ceramics or strong materials such as synthetic polymers or metals [78].

Some benefits and shortcomings of the main types of materials used for the construction of bioactive coatings are summarized in Table 3.

### 6.3. Active Moiety-Containing Coatings

Cellular active substances such as growth factors, chemokines, drugs, etc., and their application on the surface of orthopedic implants are intensively examined because these substances can effectively improve surface biocompatibility and promote osseointegration. Depending on the active moieties, these coatings can be divided into (a) drug-containing; (b) osteogenic-factor-containing; (c) immunomodulatory factors-containing and (d) antibacterial films.

#### 6.3.1. Drug-Containing Coatings

Osteoinductive drugs that accelerate bone formation and enhance implant fixation are zoledronic acid and simvastatin. They can be both loaded onto the surface of the coating or within the film. For instance, zoledronate is a long-acting bisphosphonate that was found to cause cytoskeletal alterations in osteoclasts which decreased their activity and triggered apoptosis [347]. Bilayer coating of zoledronic acid associated with CaP on Mg-Sr alloy enhanced the proliferation, osteogenic differentiation and mineralization of pre-osteoblast MC3T3-E1 cells but also inhibited osteoclast differentiation and induced apoptosis which balanced the bone remodeling process [348]. Because zoledronic acid shows numerous side effects such as osteonecrosis of the jaw, gastrointestinal irritation and impairment of renal function during systemic use [349], the administration of the drug in a controlled manner by a coating seems to have potential effectiveness.

Simvastatin is a molecular analog of HMG-CoA (3-hydroxy-3-methyl-glytaryl-coenzyme A) that was found to promote mesenchymal cell differentiation into osteoblast, downregulating osteoblast apoptosis and upregulating BMP-2 [350]. Electrohoretically deposited coatings consisting of simvastatin/gelatin nanospheres/chitosan composite on WE43 magnesium alloy were found to enhance the degradation resistance of the alloy substrate and simultaneously promoted osteogenic activity [351]. However, studies with rats proved that a high dosage of simvastatin (0.5–2.2 mg per site) may induce inflammation or even impair bone healing [352,353]. Therefore, controlled delivery and drug release in an appropriate dose are of prime importance.

#### 6.3.2. Coatings Containing Osteogenic Factors

Hormones, cytokines and growth factors such as bone morphogenetic proteins (BMPs), transforming growth factor-beta (TGF-β), vascular endothelial growth factor (VEGF) and nerve growth factor (NGF) are known to play an essential role in bone repair and are also used to construct bioactive surfaces. BMPs are widely used cytokines to confer osteo-inductivity, but their burst release can decrease the osteogenic effect [354]. That is why porous coating are expected to be suitable for this purpose. Teng and co-authors prepared porous structured coating by 3D printing and MAO and grafted BMPs onto the surface [355]. The release of BMP-2 sustained for more than 35 days and stimulated osseointegration between the implant and bone. Kim et al. loaded BMP-2 at different concentrations in a MgO and Mg(OH)_2_ layer produced by micro-arc coating on AZ31B magnesium alloy and found substantial proliferation and differentiation of osteoblast cells when BMP-2 was released continuously in a concentration of 50 ng/mL after four weeks, thus stimulating stable bone growth and bone formation [356].

When incorporating BMP-9 in thermosensitive collagen and depositing it onto porous Ti, Zhu et al. found that the thermosensitive collagen degraded slowly at 37 °C thus ensuring temperature-controlled sustained release and enhanced osteogenesis around the implant [357]. Similarly, powder-processed dopamine/gelatin/rhBMP-2 coated β-TCP films on Mg-Zn alloy facilitated cell proliferation and significantly enhanced the osteogenic differentiation of Sprague-Dawley rat bone marrow-derived mesenchymal stem cells in vitro [358]. The in vivo results in New Zealand rabbits showed strong stimulation of new bone formation and matched composite degradation and bone regeneration rate. Another coating strategy accounted for the polydopamine-mediated assembly of HAp-coated alkaline treated nanoparticles and immersion of BMP-2 onto the surface of AZ31 magnesium alloy, where the coated sample showed substantial BMSCs adhesion and proliferation and stimulated osteo-inductivity and osseointegration in the New Zealand rabbit model [359]. However, some clinical and pre-clinical side effects of BMP-2 include inflammatory and wound complications, ectopic bone, osteoclast activation and osteolysis, radiculopathy and urogenital events [360]. Therefore, more research is required to understand the long-term results and bio-functionality of conjugated coatings with BMPs.

#### 6.3.3. Immunomodulatory Factors Containing Coatings

Since the implant material is a foreign body, a series of immune responses can occur mainly from macrophage activation. Simultaneously, the formation of wear particles (debris) can also aggravate inflammatory reactions and dynamic imbalance between osteoblasts and osteoclasts which can trigger bone resorption and implant loosening [361]. Therefore, adapting the immunoreaction by incorporating immune factors to regulate immune response can have a beneficial effect on osseointegration. For example, Li et al. used spraying to deposit GO on Ti and loaded it with interleukin 4 (IL 4) [362]. During acute inflammation, IL-4 from the coating induced macrophage polarization to the Type 2 phenotype that is known to inhibit the development of inflammation. Besides weakened inflammatory response, the film also promoted osteogenesis.

Another strategy is based on immune regulation of the balance between osteoclasts and osteoblasts. Taking this into consideration, Lui et al. [363] conjugated osteogenic growth peptide (OGP) with N-acetylcysteine (NAC) and functionalized Ti substrates to examine bone metabolism balance in vitro. Their studies on RAW 264.7 cells demonstrated that the peptide-modified surfaces inhibited the cells from secreting inflammatory cytokines (IL-1β and TNF-α) and suppressed important transcription factors for osteo-clastogenesis. Simultaneously, the modified surfaces stimulated osteoblast spreading, proliferation and differentiation.

#### 6.3.4. Antibacterial Coatings

The ideal antibacterial coating should kill pathogens during the primary contact, thus preventing biofilm formation. Since implants exist in the organism for a long time, the implant surface should also provide antibacterial properties against late infections. Except for adding different metal ions such as Ag^+^, Cu^2+^ and Zn^2+^ with broad antimicrobial effects as previously discussed, some non-metallic compounds and biomolecules are also prominent candidates for the production of bactericidal coatings. For instance, iodine is found to have wide antibacterial activity without developing drug resistance [364]. Kato and Shirai deposited anodic oxide film on Ti substrate and ionized iodine was electrodeposited within the pores to achieve iodine content of 0, 20, 50, 60 and 100%, where 100% corresponded to 13 μg/cm^2^ [365]. In vitro and in vivo experiments showed a temporal pattern of rapid initial release and subsequently slow attenuation of iodine with approximately 30% of initial iodine content remaining after 1 year. Implants with iodine contents of >20% demonstrated sufficient antibacterial activity to prevent implant-related infections even after 1 year of implantation. Similarly, chlorhexidine can be absorbed on the bacteria’s surface and destroy the membrane permeability [366]. Micro- and nano-porous Ti surface prepared by alkaline and heat treatment and covalently conjugated with amino-silane was used to graft chlorhexidine via glutaraldehyde [367]. The surface containing 1 mg/mL chlorhexidine indicated the best antibacterial results together with good osteoblast compatibility. Even in the presence of bacteria, the surface displayed great potential for osteoblast adhesion at the implant-bone interface. Another antibacterial agent—dimethyl-amino-dodecyl methacrylate (DMADDM)—introduced in HAp-modified surface via polydopamine was gradually released during the first 4 weeks after implantation and exhibited both inhibition of the pathogenic bacteria growth and osteogenic differentiation [368].

The use of natural antimicrobial peptides (AMPs) represents another strategy for imparting the bactericidal properties of coatings. For example, a hybrid antibiofilm coating of immobilized antimicrobial peptide (D-GL13K) by functional linkers—elastin-like recombinamers (ELRs—was applied on a titanium surface [369]. The presence of AMPs in the hybrid coatings provided strong antibiofilm activity against mono-species and microcosm biofilm models together with excellent cytocompatibility towards primary gingival fibroblasts. Another coating consisting of polydopamine, cationic antimicrobial peptide LL-37 and phospholipid (1-palmitoyl-2-oleoyl-sn-glycero-3-phosphocholine or POPC) was deposited on MAO-modified titanium substrates [370]. The multilayered coating was found to alleviate the burst release of LL-37 in the initial phase leading to antibacterial activity against *S. aureus* and *E. coli*. LL-37 killed bacteria by blocking the expression of bacterial-related genes and stimulating immune response under controlled release of POPC. However, certain dose-dependent cytotoxicity of antimicrobial agents remains a concern.

Another recent strategy is based on photothermal therapy with near-infrared (NIR) irradiation that allows deeper tissue penetration than ultraviolet (UV) light and high selectivity. Because of the local warming effect, the biofilm or bacterial integrity can be destroyed but due to low selectivity heating may cause adverse side effects to the surrounding tissue [371]. By forming a multifunctional coating on the Ti surface of Indocyanine green (ICG) and mesoporous polydopamine Yang et al. were able to convert NIR light energy into heat to kill bacteria, while simultaneously ICG produced ROS to destroy bacteria cell walls (Figure 12) [372]. Moreover, the mesoporous polydopamine was functionalized with RGD peptide to endow the coated Ti with good cytocompatibility. After biofilm eradication, the coating still displayed osteogenetic and osteointegration potential. This strategy has the potential for remotely controlled eradication in vivo avoiding invasive treatment without side effects on the surrounding tissue. Song and co-authors modified TiO_2_ nanorods on titanium surfaces with dopamine and ferrocene (PDA-Fc) to obtain efficient antibacterial surfaces [373]. Because of ROS generation by PDA-Fc redox reactions and local temperature increase by photothermal transformation of ferrocene, synergetic and more efficient bactericidal activity of the coating was observed.

To realize an intelligent release of antibacterial moiety when the microenvironment changes, Sang et al. coated silk protein coating with gentamicin on the Ti surface [374]. The coating exhibited a faster gentamicin release rate in an acidic environment (characteristic for the first weeks after implantation) than in alkaline media. Another mechanism relies on controlled release based on the heating effect during infection. This possibility is shown by Li et al. who produced thermosensitive chitosan-glycerin-hydroxypropyl methylcellulose hydrogel (CGHH) that layered the top of simvastatin-loaded TiO_2_ nanotubes [218] and was discussed in Section 6.1.2.

Some approaches rely on the modification of the microenvironment to achieve osseointegration and bactericidal properties. For example, zeolitic imidazolate frameworks-67 (ZIF-67) coating loaded with osteogenic growth peptide prepared on TiO_2_ nanotubes was found to rapidly dissolve under an acidic environment, as during inflammation [375]. The hydrolysis of ZIF-67 nanoparticles released Co ions and formed an alkaline microenvironment that effectively kills *E. coli*, *S. aureus*, *S. mutans* and methicillin-resistant *S. aureus*. The coating was able to suppress the inflammatory response and simultaneously improved the mesenchymal stromal cell (MSCs) differentiation under an inflammatory environment. In vivo results also pointed to a strong antimicrobial and anti-inflammatory properties of the coated implants at the early stages of implantation and enhancement of bone-implant osteointegration at the late stage.

Table 4 reveals the typical production techniques used for the deposition of nitride, oxide, oxynitride, carbon-based, calcium phosphates and hydroxyapatite, bioactive glass, synthetic polymer and natural polymer bioactive coating materials on metallic implants.

## 7. Prospect and Challenges of Bioactive Coating Systems

The main issues related to the biological performance of metal implant materials are related to their poor biostability and bioactivity, low resistance to corrosion and ion release and inadequate mechanical properties. Malfunctioning of conventional metal biomaterials is frequently attributed to inflammation due to corrosion, debris formation, or microbial infections. Various deposition techniques have been used for the successful fabrication of multifunctional mechanically- and corrosion-resistant biocompatible coatings for metallic implants to enhance cell-implant interactions and reduce infections, inflammation and other post-operative complications. Chemical and physical techniques are usually used to modify metallic surfaces. Because of their numerous advantages, sol-gel deposition, plasma spraying, PVD and CVD coatings have been extensively studied while layer-by-layer and 3D printing films can achieve favorable controlled drug release profiles. However, the coating stability can vary dramatically depending on the production route and materials chosen. Lattice mismatch and high residual stresses enhance the degradation of coatings in the human body and result in poor adhesion to the substrate. Such knowledge and development of mathematical models for the calculation of degradation rate and/or adhesion are crucial for reducing the revision surgery that will lead to economic benefits.

The focus of future works should be towards an exploration of the tailored drug release properties of coatings in view of the release, controlled by different external and internal factors, of bioactive compounds. New methods to impart positive characteristics including self-healing, anti-inflammatory, antimicrobial, or controlled drug-release properties should be evaluated to improve the efficacy of biomedical coatings and metal implants. Although extensively studied recently, smart-release coatings still fail to demonstrate sufficient osteogenic capacity. Future surface modification techniques should form advanced multifunctional (anti-inflammatory, antibacterial, osteogenic) coatings integrating the merits of several coatings and offering beneficial approaches in one system that can also modify the microenvironment to modulate the immune response to promote osteogenesis. The key future development in hard tissue implants should aim at the fabrication of personalized implants where, by adapting the coating composition, including a selection of drugs, dosage and their release profiles, all individual needs of the patient can be faced. This insight will be crucial in manufacturing new implant coatings that will resolve the current issues. Therefore, more research is needed to evolve innovative surface modifications that will impart controlled antimicrobial activity, drug release, biocompatibility and wear and corrosion resistance. Further study of a combination of (bio)chemical, (bio)physical, or (bio)mechanical stimuli in one coating system can be beneficial to accelerate bone regeneration. It can be expected that the continuous development of multifunctional bioactive coatings will bring a revolutionary breakthrough in orthopedic implantable devices.

## 8. Conclusions

Applying coating on metallic implant materials can give competitive advantages because of their functionality, stability, durability and biocompatibility. However, those presenting the highest bioactivity usually suffer from inadequate mechanical or tribological properties or vice versa. For that reason, their properties are often improved by designing various composites. Combining the advantages of different production methods and coating materials led to better clinical results in long-term use compared to bare metal materials. Additionally, the unique functionalities of multi-layered, biomimetically structured or 3D printed coatings open a new horizon into the design of multifunctional coated implants. Therapeutic coatings with multi-beneficial effects such as controlled delivery of ions, drugs, proteins, growth factors, etc., and osteo-inductivity rely on combined or new methods for the production of highly efficient biomaterials and orthopedic implants. These methods offer precise control over the coating composition and structure, thus yielding the desired properties and providing all needs as per implant requirements. Thus, these coated implants lead to better clinical success rate and long-term use in contrast to uncoated metallic implants.

## Figures and Tables

**Figure 1 materials-16-00183-f001:**
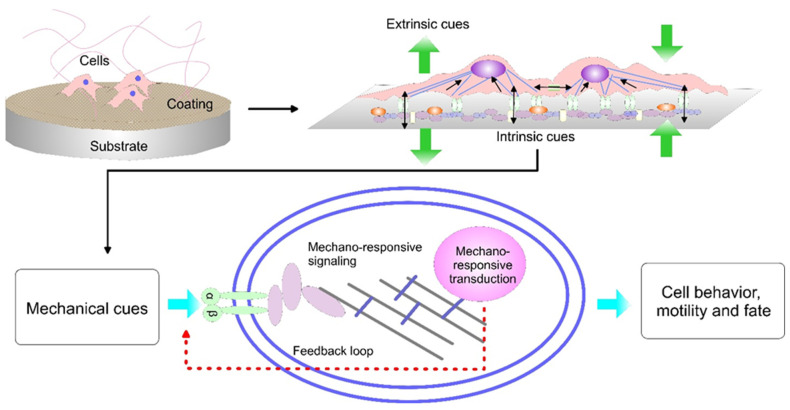
A schematic view of the influence of external mechanical forces and local (intrinsic) mechanical cues of the surrounding extracellular matrix on cell behavior of osteoblasts cultured on coated implants. The mechanical cues are transferred from the cell surface to the nucleus by the mechanisms of mechano-responsive signaling including the cytoskeleton. The mechano-responsive transduction activates downstream cell signaling cascades that determine the cell behavior, motility and fate. By a transcriptional feedback loop, the cells can change their mechano-environment by cytoskeletal remodeling and cell migration.

**Figure 2 materials-16-00183-f002:**
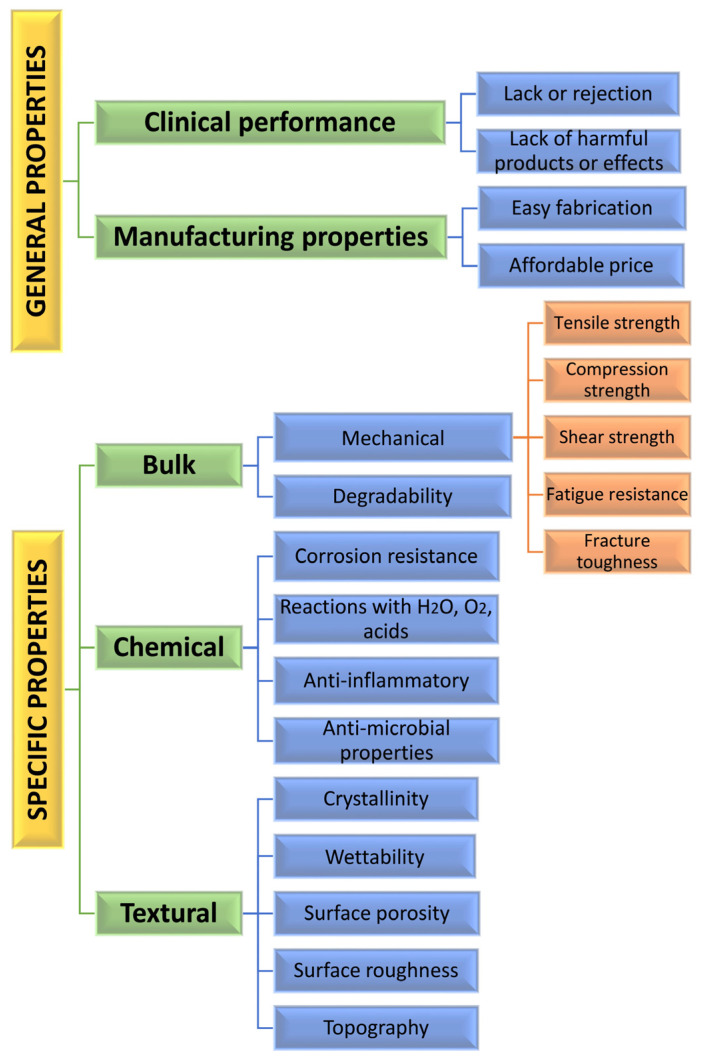
Classification of vital properties of biomaterials.

**Figure 3 materials-16-00183-f003:**
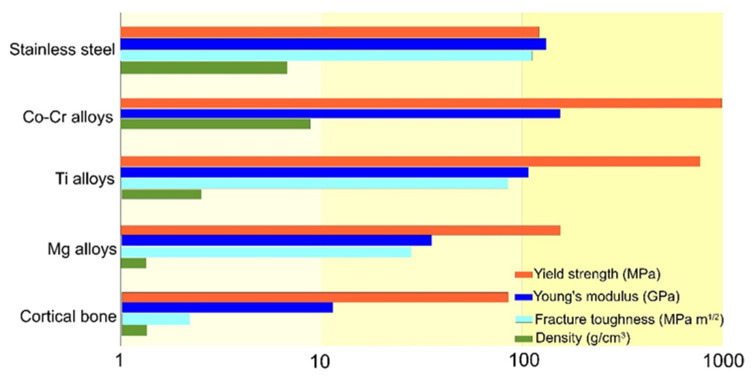
Comparison of the elastic moduli, density, yield strength and fracture toughness of different metallic biomaterials compared to those values in the cortical bone.

**Figure 4 materials-16-00183-f004:**
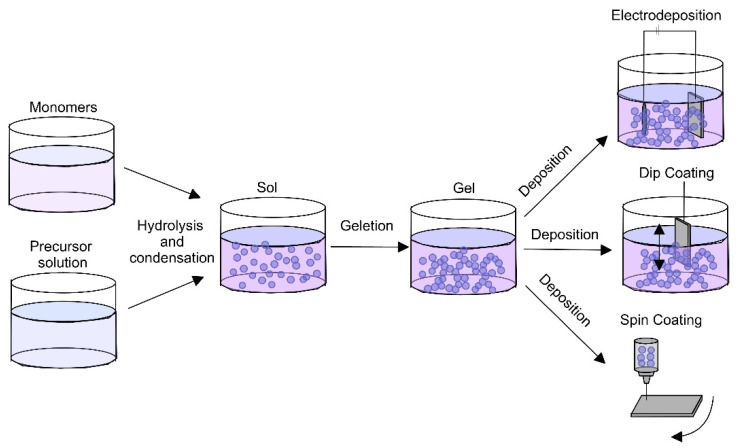
Schematic view of preparation of sol-gel preparation of coatings. Based on the type of drying method, the end product type can vary substantially.

**Figure 5 materials-16-00183-f005:**
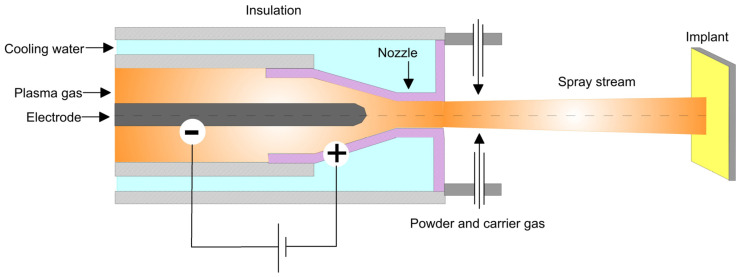
Principle of plasma generation and coating deposition on a metal substrate using plasma spraying.

**Figure 6 materials-16-00183-f006:**
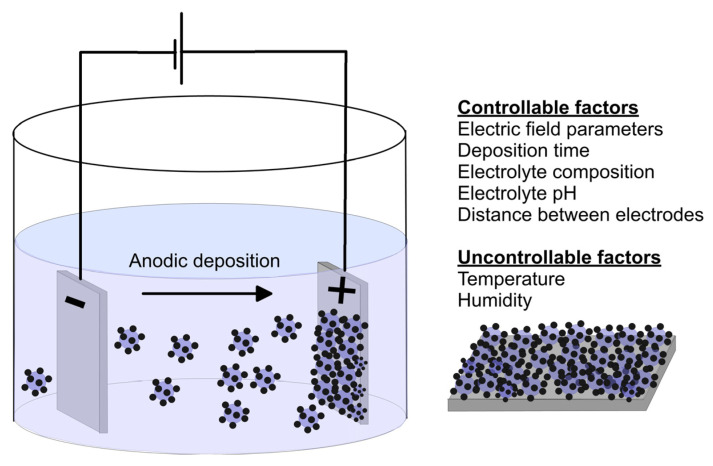
Schematic representation of the EPD process for the production of composite coatings on a metallic substrate.

**Figure 7 materials-16-00183-f007:**
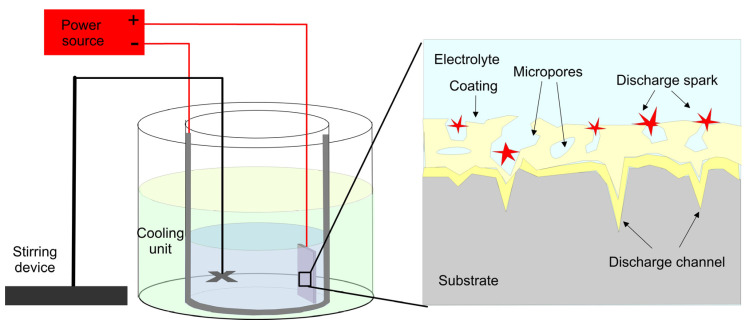
A scheme of MAO device (A) and growth model presenting the formation of MAO coating with complex pores.

**Figure 8 materials-16-00183-f008:**
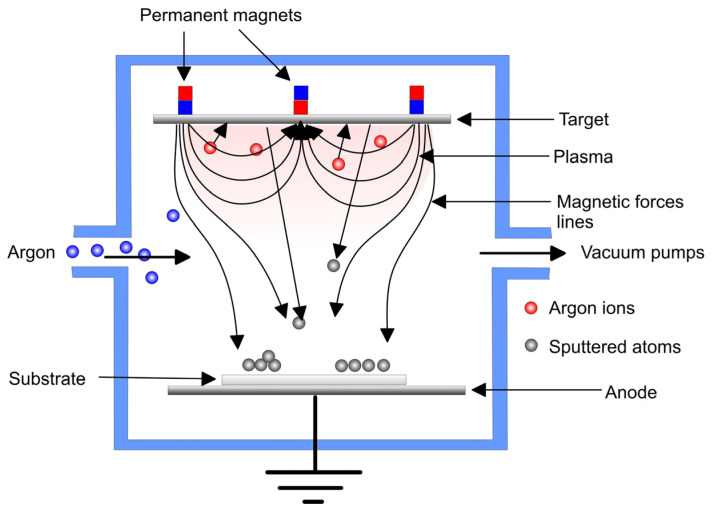
Schematic representation of magnetron sputtering PVD process for deposition of thin films.

**Figure 9 materials-16-00183-f009:**
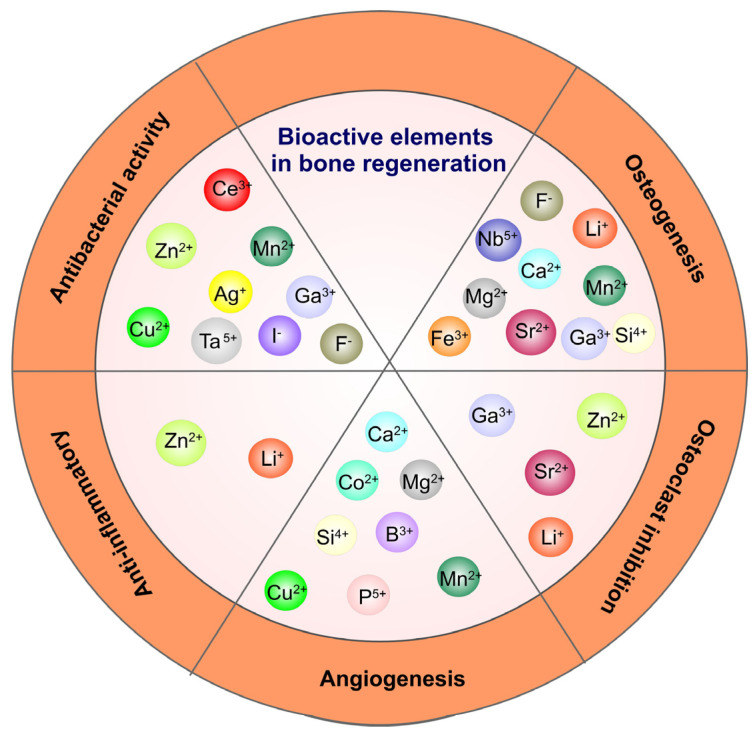
The therapeutic impact of some ions with angiogenic, osteogenic, anti-inflammatory and anti-bacterial activity.

**Figure 10 materials-16-00183-f010:**
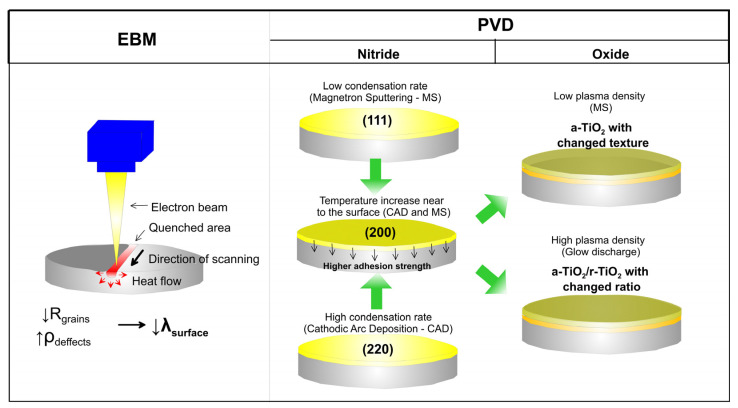
Schematic representation of the changes occurring after EBT of the metal surface and during deposition of the nitride/oxide coatings deposited by different methods.

**Figure 11 materials-16-00183-f011:**
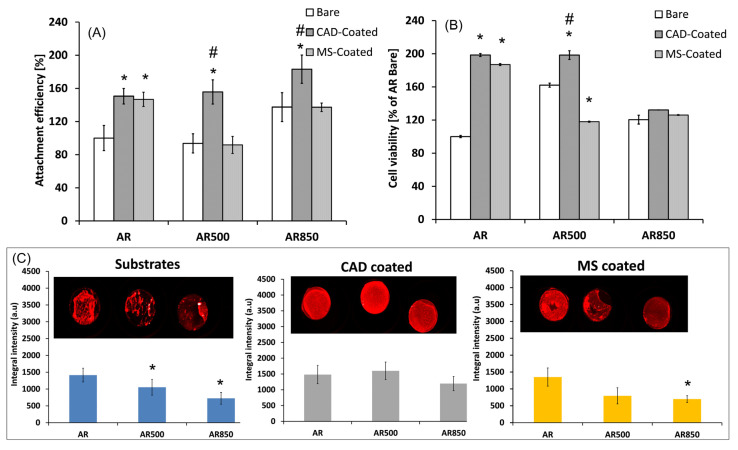
Cell adhesion after 6-h residence time. (**A**), cell viability for 24 h (**B**) and bone mineralization activity of MG63 osteoblasts after 31 days of incubation (**C**) with uncoated and TiN/TiO_2_ coated, polished and EBM samples of Ti64 in as-received (AR) state. Three independent studies of the cell culture samples were performed. Results are averaged and the standard deviation is indicated. * *p* < 0.05 compared to the polished uncoated AR sample; # *p* < 0.05 compared to the MS coated sample in the respective series.

**Figure 12 materials-16-00183-f012:**
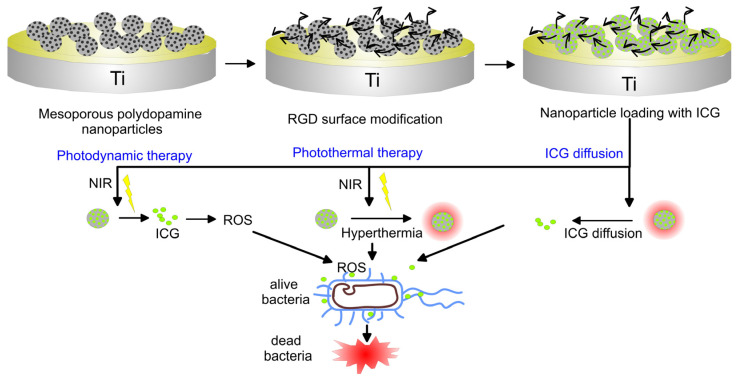
Schematic illustration revealing the coating construction and the elimination process of bacteria biofilm through NIR light triggering remote photodynamic and photothermal synergetic treatment.

**Table 1 materials-16-00183-t001:** Advantages and disadvantages of the coating methods used for deposition of bioactive coatings on metals and alloys developed for medical application.

Classification	Method	Advantages	Disadvantages	Ref.
Chemical	Sol-gel	Simplicity, cost-effectiveness, easy incorporation of active ingredients, various film thicknesses at complex shapes	Long processing time, low adhesion, need for post-sintering, the occurrence of cracks	[81,82,83,84,87]
	Electroless deposition	Simple, low-temperature process, suitable for complex geometries, excellent thickness uniformity	A lengthy process, microcavities and pitting in the deposits	[88,89,90]
	Biomimetic deposition	Low-temperature process, incorporation of osteogenic agents	Need for initial pretreatment, long processing time, poor bonding strength	[91,92,93]
	Spray pyrolysis	Simple, cost-effective, low setting-up costs	Need for post-deposition annealing	[99]
Thermo-chemical	Thermal spraying	Microrough coating surfaces, cost-effective technique, high deposition rate	High-temperature process; porous, inhomogeneous and amorphous films, low adhesion strength and prone to coating spallation	[80,93]
	Hydrothermal	Simplicity, cost-effectiveness, controllable morphology, high purity	Long experimental time	[112,170]
	CVD	Uniform film thickness, controlled properties of the deposits, functionalizing of multidimensional surfaces	limited volatile components, generation of toxic by-products, use of toxic precursors, requires expensive equipment	[114,115]
Electrochemical	Electrophoretic deposition	Short processing time, porous coatings with controllable thickness, incorporation of bioactive ingredients	Low adhesion strength, need for sintering, the occurrence of cracks, difficult scalability	[120,121,122,123]
	Electrochemical deposition	Low-temperature process, controlled coating structure and morphology, relatively low costs	inadequate bonding strength; development of stresses in the coatings; non-homogeneous and non-stoichiometric surfaces	[125,127,128,129]
	Anodization	Variable nanotube diameters, drug-loading capacity and controlled release	Annealing is required, low mechanical stability, the release of toxic ions	[130,131,132]
	Micro-arc oxidation	Highly microporous oxide layers incorporating different compounds from the electrolyte, good adhesion, simplicity, environmentally friendly process	Low crystallinity, cracks formation, need for subsequent treatment, corrosion instability	[134,138,139]
Physical and electrophysical	PVD	Environmentally friendly process, the coatings have superior mechanical biocompatible properties, the deposits can be applied to temperature-sensitive substrates	High equipment costs, inability to incorporate biomolecules	[140]
	Ion implantation	Strong adhesive bonding	Limited sample size, need for rotation, amorphous film formation, high costs	[141,150]
	Electrospinning	Simple and flexible technique, production of fibers with various diameters incorporating particles or drugs, slow release of bioactive components, controlled internal porosity, scale-up potential	Low mechanical and bonding strength	[157]
	Laser cladding	High deposition rate, good coating adhesion, increased surface roughness	Formation of pores and cracks because of different thermal expansion coefficients of the substrate and coating	[152,153]
3D printing	fused deposition modeling, semi-solid extrusion	Flexibility in terms of materials used, obtaining complex geometric shapes, time efficacy, low production costs	Coating swelling, low bonding strength, low long-term instability, applied heat to the drug during manufacturing	[160,162,163]

**Table 2 materials-16-00183-t002:** Requirements for HAp coatings for biomedical application [291].

Property	Specification
Ca/P ratio	1.67–1.76
Heavy metals	<50 ppm
Density	2.98 g/cm^3^
Crystallinity	>62%
Thickness	5–70 μm
Abrasion	Mass loss < 65 mg at 100 cycles
Tensile strength	>50.8 MPa
Shear strength	>22 MPa

**Table 3 materials-16-00183-t003:** Pros and cons of various bioactive coatings applied for hard tissue applications.

Coating Material	Benefits	Shortcomings	Ref.
Metal nitrides (TiN, ZrN, TaN, SiN) and oxynitrides (TiN_x_O_y,_ TiN/TiO_2_, TiON, ZrON)	Acceptable adhesion to metal substrates, high wear and corrosion resistance, biocompatibility	High hardness, premature coating failure, formation of flakes	[175,177,179,183,196,197]
Metal oxides (TiO_2_, Ta_2_O_5_)	Good mechanical properties, bioactivity, antibacterial and catalytic activity, long-term stability under the photo- and chemical corrosion	Brittleness, low fracture toughness	[57,200,201]
Carbon-based (DLC, nanodiamonds, graphene, GO)	Biocompatibility, stability, good mechanical properties, low coefficient of friction, superior electrochemical properties, antimicrobial properties	Single-layered DLC films suffer from high internal stress, delamination in an aqueous environment, low toughness, high sensitivity to ambient conditions; Nanodiamonds and graphene display hydrophobicity; GO nanomaterials indicated the generation of ROS, DNA damage and mitochondrial disturbance;	[233,234,237,238,239,240,247,250,251,270]
Calcium phosphates and hydroxyapatite	Exceptional biocompatibility, osteo-inductivity, osteoconductive, bioactivity	Very brittle, high stiffness, low flexibility, high solubility in aqueous media	[78,271,273]
Bioactive glasses	Class A bioactivity, no toxic effects	Semicrystalline or amorphous structure, high brittleness, low fracture toughness, interfacial delamination, need for sintering to achieve adequate adhesion to a metallic substrate	[292,297,302,346]
Synthetic polymers (PMMA, PPy, PU, PCL, PGMA)	Inexpensive, biodegradable (PCL, PGMA), chemically stable, good tensile properties and flexural rigidity	Low mechanical properties, hydrophobicity, insolubility (PMMA, PU, PPy), inflammatory reactions, inadequate degradation rate, deteriorate bone cell adhesion	[57,58,309,311,315,320]
Natural polymers (chitosan, silk fibroin, dopamine)	Good cell adhesion, high cell affinity	Low adhesion to metals, low durability, insufficient mechanical properties	[326,328,336]
ECM proteins/cell coatings	Enhance tissue regeneration of bone, tendon, ligaments and vascular and connective tissue, promoting cell adhesion, probiotic activity	Low mechanical strength, difficult sterilization, rapid degradation	[78,339,341,345]

**Table 4 materials-16-00183-t004:** Typical production techniques used for the deposition of various bioactive coating materials on metallic implants.

Coating Material	Type	Deposition Technique	Ref.
Nitrides	TiN on Ti6Al4V	Ion implantation	[147]
	TiN and “soft” Ti_4_N_3−x_ on Ti6Al4V alloy	DC magnetron sputtering	[181]
	TiN coating on Ti20Nb13Zr (TNZ) alloy	Cathodic arc PVD	[182]
	Copper-doped TiN (TiCuN) deposited on 316L SS	Axial magnetic field enhanced arc ion plating	[186]
Oxides	ZnO and ZnO/Ag on Mg-Ca alloy	Electroless deposition	[89]
	TiO_2_ layers on Ti sheets and TiO_2_ nanotubes	Atomic layer deposition	[116]
	Tantalum oxide on Mg alloy	Reactive magnetron sputtering	[203]
	Doped-TiO_2_ coatings	MAO (PEO)	[206,208,209,211]
	TiO_2_ nanotubes on pure Ti	Anodization	[207]
	TiO_2_ layer embedding silver (Ag) and zinc (Zn) nanoparticles	3D printing	[212]
	Ag-doped TiO_2_ coatings on Ti	Sol-gel	[213]
	Collagen/polydopamine/TiO_2_ coatings on Ti implants	MAO and hydrothermal treatment	[216]
Oxynitrides	TiN/TiO_2_ coatings on Ti6Al4V	Cathodic arc deposition (CAD) and glow discharge oxidation	[225]
	TiON and ZrON films on SS 316L	Magnetron sputtering	[229]
Carbon-based	Hybrid DLC coatings incorporating TiO_2_ nanoparticles on AISI 316	plasma-enhanced CVD	[246]
	Nanocrystalline and microcrystalline diamonds on Mo substrates	Hot filament CVD	[257]
	Modified ultra-nanocrystalline diamond coatings on Ti	Microwaved plasma-assisted CVD and electron-beam evaporation	[253]
	DLC with Zr-containing interlayers	Unbalanced magnetron sputtering	[242]
	Mg-functionalized GO coating on Ti6Al4V	Electrophoretic deposition	[262]
	Graphene on pure Ti	Liquid-free technique	[268]
	Apatite-nanodiamond coating on SS	Electrodeposition	[256]
	GO loaded with interleukin 4 on Ti	Spraying	[362]
Calcium phosphates and hydroxyapatite	Ultra-porous HAp on Ti alloy	Spray pyrolysis	[102]
	HAp-graphene on Ti substrates	Cold spraying	[107]
	HAp on Ti substrates	Hot isostatic pressing	[109]
	CaP layer on Mg alloy	Hydrothermal crystallization	[110]
	Si-HAp coating on Mg-5Zn-0.3Ca alloy	Pulse electrodeposition	[125]
	HAp coatings on WE43 Mg alloy	EPD and pulse laser deposition (PLD)	[276]
	HAp with TiO_2_ on Ti6Al4V alloy	High-Velocity Oxygen Fuel (HVOF) spraying	[277]
	Ca-Sr-P coatings on Mg alloy	Chemical immersion method	[287]
	Ag-F-HAp coatings on Ti substrate	Sol-gel method	[288]
	Ce-doped HAp/collagen coatings on Ti	Biomimetic deposition	[290]
Bioactive glasses	Zirconia-incorporated bioactive glass films on pure Ti substrates	Spray pyrolysis	[100]
	Bioactive glass onto ultra-fine-grained Ti substrates	Laser cladding	[152]
	CaO-MgO-SiO2-based bioactive glass-ceramic on Ti6-Al-4V alloy	Atmospheric plasma spraying	[294]
	Bioactive glass nanorods of 45S5 integrated with rGO sheets	Sol-gel deposition	[299]
	Bioactive glass-rGO hybrid thin on TiO_2_ nanotubes	Electrophoretic deposition	[300]
	Bioglass coatings reinforced with Ti on Ti substrate	Laser Engineered Net Shaping (LENS)	[301]
Synthetic polymers	Poly(sodium styrene sulfonate) on Ti	Wet-chemical method	[96]
	PU coating with graphene and β-TCP on Ti implants	Dip coating	[309]
	PPy/ZnO composite coating on Mg alloy	Electrochemical synthesis	[316]
	PCL/gelatin coatings on 316 SS	Electrospinning	[320]
	PEEK-HAp composite coating on 316 SS	Electrophoretic deposition	[318]
	PGMA coupled with quaternized polyethyleneimine and alendronate on Ti implants	Immersion treatment	[321]
Polymeric gels (natural polymers)	Methoxyl pectin and xanthan incorporating indomethacin coatings on SS	Sol-gel	[86]
	Chitosan over porous oxide layer on Ti6Al4V	Electrodeposition	[126]
	Chitosan-Mg composite coating on Mg-Gd alloy	Dip coating	[323]
	Carboxymethyl chitosan coating on AZ31D alloy	Immersion treatment	[324]
	Chitosan/hyaluronic acid coating on Ti substrate	Layer-by-layer synthesis	[325]
	Silk fibroin layer on HAp/Mg(OH)_2_ coating on WE43 magnesium alloy	Spin coating	[337]
	Simvastatin/gelatin nanospheres/chitosan composite on WE43 magnesium alloy	Electrophoretic deposition	[351]

## Data Availability

Not applicable.

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
