# Peer review of "Advances in Multifunctional Bioactive Coatings for Metallic Bone Implants"

_materials, 2022, doi:10.3390/ma16010183_

Round 1

Reviewer 1 Report

The article presents a comprehensive review on the development attained in the bioactive coatings for metallic bone implants area. The paper is well written and organized. The references used in this review are ~55% from the last 5 years and ~30% from the last 3 years (2020-2022). The English must be carefully checked (ex. “and is” replace by “being”). This work can be considered for publication after addressing the following comments, which help the authors to improve their manuscript and the readers through their reading.

i) Define all acronyms before using them, ex: RGD, PHSRN, KRSR, DGEA, BMP2-CBD, rh-BMP2, PVA, PAA, PEEK, ROS, etc.

ii) In order to emphasize the particular features of each technique, at the end of “Coating methods” section, the authors must summarize in a table the techniques involved in the preparation of bioactive coatings mentioning their advantages and disadvantages. In the Table, for each technique, the authors must provide the corresponding references.

iii) In order to emphasize the specific characteristics of each material, at the end of “Bioactive coatings” section, the authors must summarize in a table the materials used as bioactive coatings mentioning their advantages and disadvantages. In the Table, for each material, the authors must provide the corresponding references.

iv) In order to emphasize the link between the materials and the techniques described in the review, before “Prospect and challenges of bioactive coating systems” section, the authors must summarize in a table the materials used as bioactive coatings and the techniques usually used in the deposition of each material together to the corresponding references.

v) The conclusions are too vague. The authors must rewrite concisely this section based on the information (techniques and materials used in the development of the bioactive coatings for metallic bone implants) presented in the paper.

The authors must provide a point-by-point response to the reviewer comments.

Reviewer 2 Report

Interesting, well written and comprehensive review paper.

I only recommend few minor improvements:

1. Abstract, line 11: saying that metallic implants are inert is misleading to some extent, also considering what you write few lines below (corrosion, infections induced by the metal ion released etc.); please revise.

2. This recent review on bioactive coatings deserves to cited:

Bioceramic coatings on metallic implants: an overview. Ceramics International 2022;48:8987-9005

3. Antibacterial coatings produced by sputtering were not cited, you could consider this paper:

Silver nanocluster/silica composite coatings obtained by sputtering for antibacterial applications. IOP Conf Series: Mater Sci Eng 2012;40:012037

4. Some tables summarizing the main findings cited in the review would be helpful to Readers. Perhaps, you could prepare one Table per each section/type of coating.

Reviewer 3 Report

This paper systematically reviewed the bioactive coatings on implant surfaces, aiming at concluding the research and development of coatings on metal implants in the biomedical field. The bioactive materials and their properties were emphatically discussed, and the possibility of further improving the performance of bioactive ceramic coatings by adding and controlling the release of drugs into the coatings was preliminarily discussed. The authors have further selected the collected article on the basis of relevant scientific theories and research needs of metal implant surface coatings in the biomedical field, and a detailed and systematic review of "research problems, objectives, methods, results and discussions" and their "existing problems", "research deficiencies" and "problems not yet raised" in each article.

Although the authors' review of the relevant article establishes an important framework system for the development of research on coatings of metal implants in the biomedical field. However, it is important to realize that such a significative article review is not only for researchers in the relevant field to read, but is more primarily presented to other readers. Therefore, before accepting it, the following points are suggested:

Comment 1: Improve the readability of the paper. For example, the author should make it easier for the reader to understand the numerous abbreviations used in the paper.

Comment 2: Currently, laser cladding is commonly used as an advanced surface modification technique to improve the wear resistance, corrosion resistance, or biological properties of material surfaces. Therefore, it is suggested that the authors review the current research progress in the preparation of bioactive coatings by laser cladding to further improve the research framework of metal implant surface coatings in the biomedical field.

Comment 3: The authors point out that bioactive coatings have defects, such as poor mechanical strength and low bond strength. However, the current preparation of bioactive coatings also has a large number of process problems. For example, the nature of bioactive material substances determines their easy decomposition in high-temperature environments, which leads to the tendency of bioactive coatings prepared by processes that require high-temperature processing to lose their original functions. Therefore, it is suggested that the authors are needed to easily review the process difficulties in bioactive coating preparation methods and their possible solutions.
